# A Comprehensive Review of Architecture, Communication, and Cybersecurity in Networked Microgrid Systems

**Ahmed Aghmadi** [ID], **Hossam Hussein** [ID], **Ketulkumar Hitesh Polara and Osama Mohammed** *[ID]

Energy Systems Research Laboratory, Department of Electrical and Computer Engineering,
Florida International University, Miami, FL 33174, USA; aaghm001@fiu.edu (A.A.); hhuss013@fiu.edu (H.H.);
kpola009@fiu.edu (K.H.P.)
* Correspondence: mohammed@fiu.edu

**Abstract:** Networked microgrids (NMGs) are developing as a viable approach for integrating an expanding number of distributed energy resources (DERs) while improving energy system performance. NMGs, as compared to typical power systems, are constructed of many linked microgrids that can function independently or as part of a more extensive network. This allows NMGs to be more flexible, dependable, and efficient. The present study comprehensively investigates architecture, communication, and cybersecurity issues in NMGs. This comprehensive study examines various aspects related to networked microgrids (NMGs). It explores the architecture of NMGs, including control techniques, protection, standards, and the challenges associated with their adoption. Additionally, it investigates communication in NMGs, focusing on the technologies, protocols, and the impact of communication on the functioning of these systems. Furthermore, this study addresses cybersecurity challenges specific to NMGs, such as diverse cyberattack types, detection and mitigation strategies, and the importance of awareness training. The findings of this study offer valuable insights for NMG researchers and practitioners, emphasizing critical aspects that must be considered to ensure the safe and efficient operation of these systems.

**Keywords:** networked microgrids (NMGs); distributed energy resources (DERs); architecture and control methods; communication; cybersecurity; standards; regulations



## 1. Introduction

In recent years, the growing demand for reliable and sustainable energy systems has increased interest in microgrids. A microgrid is a small-scale, localized power system that combines various energy generation, storage, and load control methods. It runs attached to or separate from the electrical grid and offers a dependable, long-lasting, and affordable energy source to a particular user, building, or industry. Microgrids are becoming increasingly common because of the need to lessen reliance on centralized power networks and rising worries about energy security. Microgrids may combine several renewable energy sources to offer a renewable energy balance. The central controller system of a microgrid may manage the production of several energy sources in addition to managing the energy requirements of linked loads. As a consequence, energy usage and expenses are decreased [1].

With the incorporation of microgrids into the power sector, many operational advantages are available over conventional grids, including [2]:

(1) Reliability and Resilience of Power Networks: Microgrids have the potential to disconnect from the primary grid and operate in "island mode," utilizing their local energy generation and storage systems, which allows them to continue working in times of emergency or power outages.

(2) Increased Renewable Energy Integration: Microgrids are a more sustainable and environmentally friendly choice for energy generation since they can include many renewable energy sources, such as solar panels and wind turbines.

(3) Improved Energy Efficiency: The central management system of a microgrid regulates the energy demand of the connected loads and coordinates the output of the various energy sources, resulting in the more effective use of energy resources and lower energy expenditures.

(4) Enhanced Grid Stability: Microgrids can complement the primary electrical grid by supplying extra capacity during high demand and lessening the burden on the central grid, boosting the stability and reliability of the entire power system.

(5) Local Control and Ownership: Microgrids are commonly owned by local communities or facilities, giving them more control over their energy supply and reducing their dependence on centralized power systems.

It is becoming increasingly apparent that AC, DC, and hybrid AC/DC MGs are promising platforms for the further development of power system architectures, and the choice of microgrid architecture will depend on the specific needs and requirements of the community or facility that it serves. More MG variants have emerged recently with the further development of power electronics and energy storage system (ESS) technologies. For example, clustering MGs [3] involves grouping microgrids based on proximity or shared resources. The primary objective of clustering microgrids is to facilitate collaboration and optimize performance among the grouped microgrids. This approach can improve resource sharing, load balancing, and system efficiency.

In contrast, a multi-energy microgrid [4] integrates various energy carriers and technologies to enable a diversified and flexible energy supply. By incorporating different energy sources and technologies, multi-energy microgrids can enhance resilience, facilitate optimal resource allocation, and accommodate various energy demands. Networked microgrids (NMGs) [5] are interconnected microgrids that exchange power and information. This interconnectedness enables coordination and cooperation among the microgrids, improving efficiency and resilience at the network level. NMGs can leverage component synergies, optimize resource allocation, and enhance system performance. They offer benefits such as increased reliability, improved load management, and the efficient incorporation of distributed energy resources. Additionally, shipboard/aviation MGs [6] are specifically designed to cater to maritime or aviation applications' unique energy requirements and operational constraints. Shipboard/aviation microgrids focus on ensuring a reliable power supply, efficient energy utilization, and enhanced system resilience in challenging environments such as ships and aircraft. These microgrids often involve specialized energy management systems and technologies tailored to meet the specific demands of onboard power systems. These advancements have the potential to significantly improve the efficiency of onshore distribution systems [7], and reducing emissions from offshore and aerospace microgrids, such as all-electric ships (AESs) [8,9], is becoming a priority. The shift toward integrating multiple energy sources has resulted in the creation of multi-micro-energy microgrids [10].

Microgrids operate in two modes: grid-connected and standalone. In grid-connected mode (Figure 1a), the microgrid remains connected to the primary grid, importing or exporting energy as needed. If there is a failure in the primary grid connection, the microgrid switches to standalone mode (Figure 1b) and continues to power critical loads. This can be achieved by disconnecting the entire microgrid from the primary grid or by disconnecting specific feeders within the microgrid. If the whole microgrid is disconnected, all micro-sources must supply energy to all loads in all feeders. If only particular feeders are disconnected, they will provide power to the loads while others ride out the disruption [11].

The energy management functions in microgrids ensure a microgrid's efficiency and economic activities in different scenarios based on multiple factors, such as the device state, the predicted load and weather, and the power of distributed energy resources (DERs). An EMS may connect and manage the energy output of DERs, energy storage systems, and energy transfers [12]. Many models of energy management systems have been developed with drawbacks, either technical, economic, or environmental [13–18].

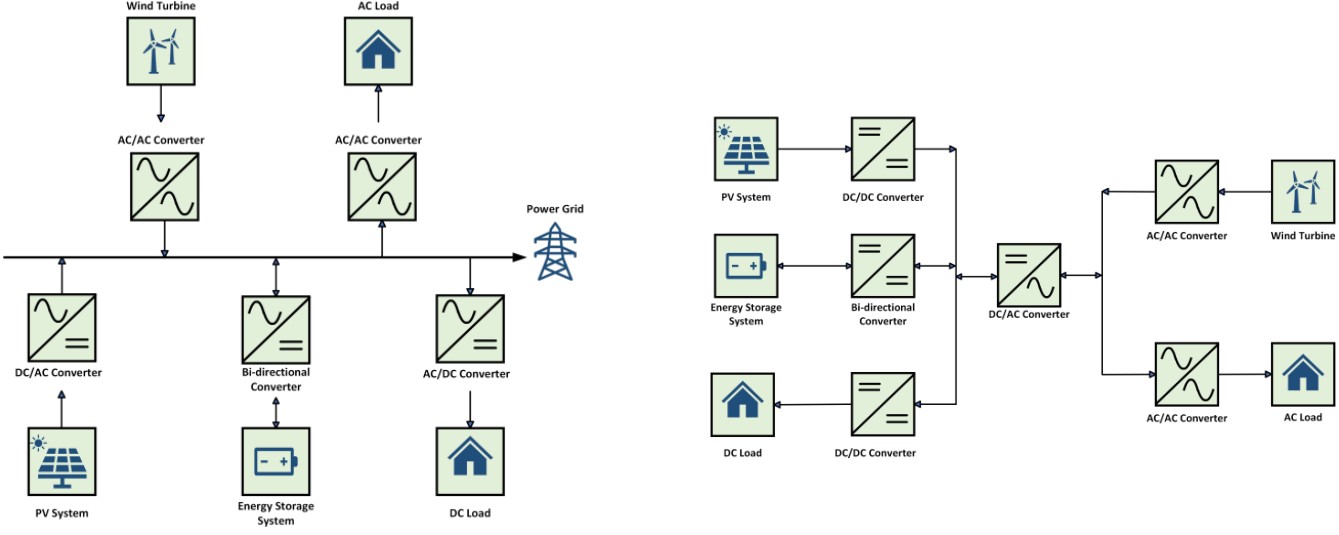

**(a) Grid-Connected Mode**　　　　　　　**(b) Standalone Mode**

**Figure 1.** Microgrid operation modes: (**a**) grid-connected mode; (**b**) standalone mode.

Networked microgrid systems can achieve flexibility and complementarity among various energy resources by combining the improved energy management of MGs [19,20]. With the aid of different energy conversion technologies [21,22], it is advantageous for MGs that produce surplus electricity to trade it with other MGs that seem to be short on electricity and would otherwise need to buy energy from the main grid to reduce transmission losses and prevent grid outages. Two operation state modes can be envisioned in networked microgrid systems. These are the two potential modes of operation [23]:

- Normal operation mode: when the networked microgrid system is connected to the primary power distribution network.
- Emergency operation mode: when the networked microgrid system is functioning independently or when, after a power outage, it assists in restoring service by starting up on its own.

The challenge of controlling networked microgrid systems lies in the use of separate controllers with varying degrees of independence and the ability to communicate and implement control strategies. As the systems become larger and more complex, partially decentralized approaches are necessary, and therefore, networked microgrid systems are managed by using more flexible control structures. A central controller collects information from various devices and sets rules for local controllers. However, managing all this information through central management can become overwhelming, making it difficult to effectively control the system, especially during islanded or standalone mode [24].

The networked microgrid system has a self-healing function that enables it to withstand and recover from breakdowns or failures, such as those caused by natural disasters, in a geographic area [25–29]. This helps protect individual microgrids (MGs) from damage and allows them to continue functioning smoothly by exchanging energy and resources among interconnected microgrids, even in a failure or disaster. This helps to protect the microgrid and maintain steady operation [26,27]. Ensuring stability and reliability in networked microgrid systems (NMGs) remains challenging despite the advantages. This is because NMGs heavily rely on renewable energy (RE), which tends to be unstable and intermittent, resulting in more fluctuations and variability in time and space. Finally, ensuring information security and protecting the system from potential cyber threats is also a challenge in NMGs, as the system is highly connected and reliant on communication networks. These challenges highlight the need for effective and efficient solutions that can address the unique difficulties of NMGs and ensure stable and reliable operation.

One of the main differences between this article and other review papers is that it provides a comprehensive and detailed overview of all aspects of networked microgrid systems, whether architectures and controls, communication technologies and protocols, and cybersecurity. The authors of this paper are aware of a few other review papers published on networked microgrid systems in the literature. These papers focus on aspects such as architecture, communication, and cybersecurity. In contrast, this review paper provides a comprehensive and detailed overview of networked microgrid systems. As a result, it serves as a crucial resource for scholars and professionals interested in discovering more about this recent technology.

The main contributions of this work can be summarized as follows:

1.  We provide a comprehensive overview of networked microgrids in terms of the architectures, control strategies, challenges and benefits, and standards and regulations of NMGs.
2.  We present a detailed review of the most widely used communication technologies and protocols in NMGs, including the requirements for appropriate functioning and their impact.
3.  We provide a critical review of cybersecurity attacks on NMGs, including case studies reported in the literature.
4.  We also provide a detailed review of mitigation techniques to prevent cyberattacks.

The rest of this article is organized as follows. Section 2 reviews the networked microgrid architecture, control structures, and strategies. Furthermore, challenges and benefits are discussed, along with the standards and regulations of NMGs. Section 3 reviews communication systems' general requirements and challenges and investigates their impact on NMGs. In addition, communication technologies and protocols are highlighted. Section 4 reviews the types of cyberattacks and threats to NMG systems and detection and mitigation techniques. Finally, the conclusion of this article is in Section 5.

## 2. Networked Microgrid Architectures, Controls, and Standards/Regulations

Microgrids have traditionally been used to power standalone loads unrelated to the primary grid during grid outages and emergencies. Gradually, they revolutionized as a response to the growing integration of renewable energy. The MG structure includes uncontrollable sources such as solar and wind generators, dispatchable sources such as diesel and natural gas generators, and different varieties of thermal, mechanical, electrical, chemical, and electrochemical energy storage systems (ESSs), including flywheels, supercapacitors, and batteries [30]. The concept of networked microgrids (NMGs) has recently attracted growing attention due to their advantages, novelty, effectiveness, and robustness over conventional individual microgrids.

NMGs are an amalgamation of physically interconnected nearby microgrids, as shown in Figure 2, while functionally interoperable. According to the NMG paradigm, multiple microgrids are linked to distribution feeders via points of common coupling (PCCs), either with fixed electrical borders or dynamic boundaries created by distributed or centralized intelligence [31]. The main purpose of these types of networking is to ensure mutual power sharing between multiple microgrids, which will result in operational cost reductions, improve overall system reliability and resiliency, and enable fast service restoration after a fault or power deficiency event. Additionally, NMGs enable the efficient utilization of renewable energy resources (RESs) in islanded or grid-connected modes of operation, which can significantly minimize carbon dioxide emissions and strains on the main grid in grid-connected processes [32]. However, many challenges come with the widespread use of NMGs in terms of stability, control, protection coordination, and security [33]. This section goes through the different architectures of NMGs, the various control strategies and techniques, and the operational aspects of NMGs and ends with the technical benefits and challenges of massive expansion in the implementation of NMGs.

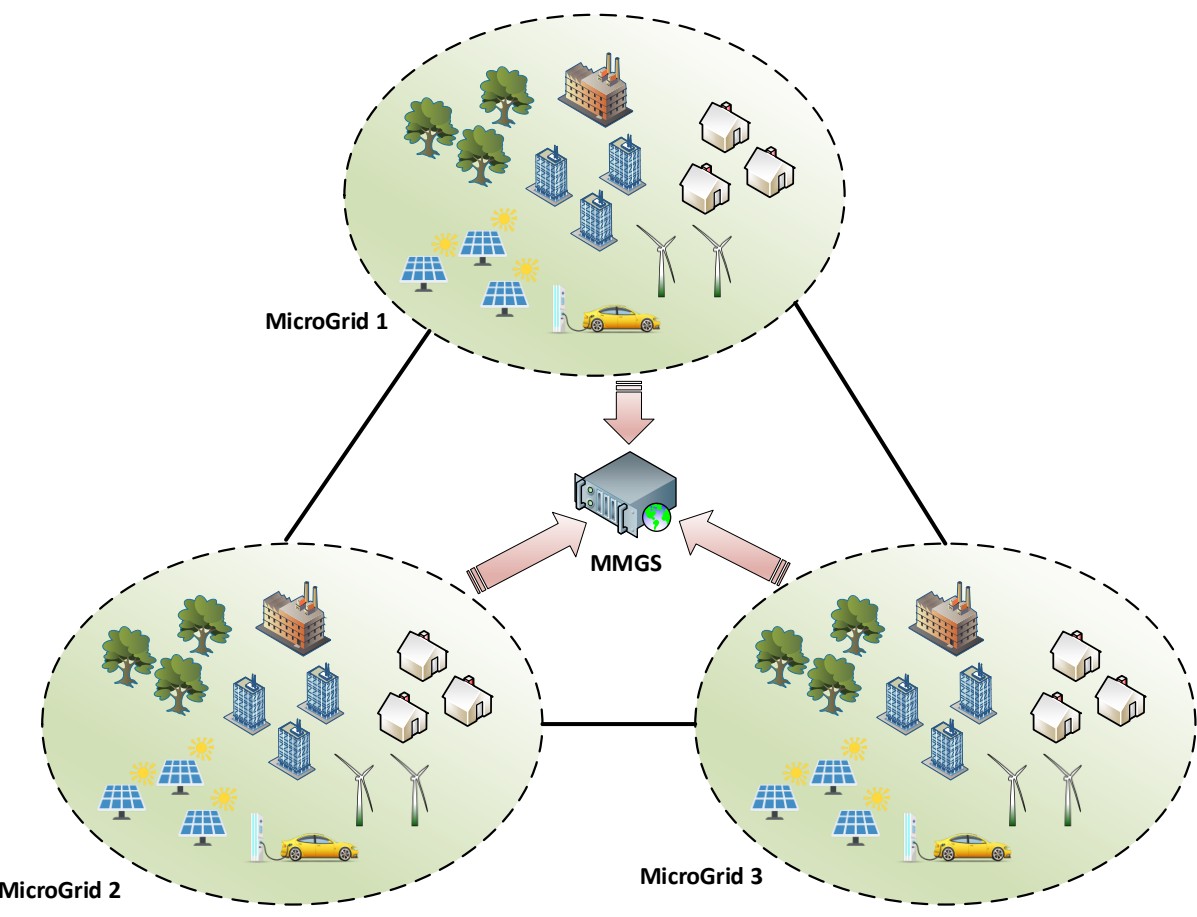

**Figure 2.** Networked microgrid overview.

### 2.1. Architecture of NMGs

Practically, restricting the design of NMGs to two or three categories makes no sense when the main goal is to increase the energy exchange capabilities among them. The interconnection between clustering microgrids and the distribution network provides numerous benefits in terms of economic optimization, resiliency, and reliability. Microgrids are linked and ultimately create NMGs to interchange electric power at various time scales, such as enhancing transient stability within seconds, optimizing the power flow every few minutes or hours, and offering continuous power to customers impacted by outages. Since there is a vast range of interconnections, NMGs can be categorized according to their physical connections' operational flexibility [31].

The most typical method of establishing an NMG system is a radial or star topology, as shown in Figure 3a. This might be accomplished by connecting the MGs directly to the main grid, with no direct connections between MGs. In the event of a disruption in the upstream network, MGs can detach from the main grid and enter islanded mode. In addition, if the main control and communication system can accommodate more MGs, expanding and attaching them to the design above is quite simple. However, this architecture fails to ensure the reliability of the network or the main grid since there is only one feeder from the grid with the MGs, which will be stressed or overloaded on some occasions [32,34,35]. Increasing the overall system's reliability and shared power for the architecture above can be achieved by connecting each MG directly to its neighbors, forming a daisy-chain topology, which can also be called parallel MGs on a single feeder, as shown in Figure 3b. Stronger interactions between MGs result in extra network restrictions and linked energy plans for optimal scheduling [36]. Furthermore, each MG must operate independently, determining its operation schedule according to its load profiles and based on non-critical information from peer-to-peer communications with neighboring MGs [37]. The amount of

power shared between NMGs can be expanded by modifying the radial system to a mesh topology by adding a direct connection between all the MGs, as shown in Figure 3c. Table 1 illustrates NMG architectures reported in the literature.

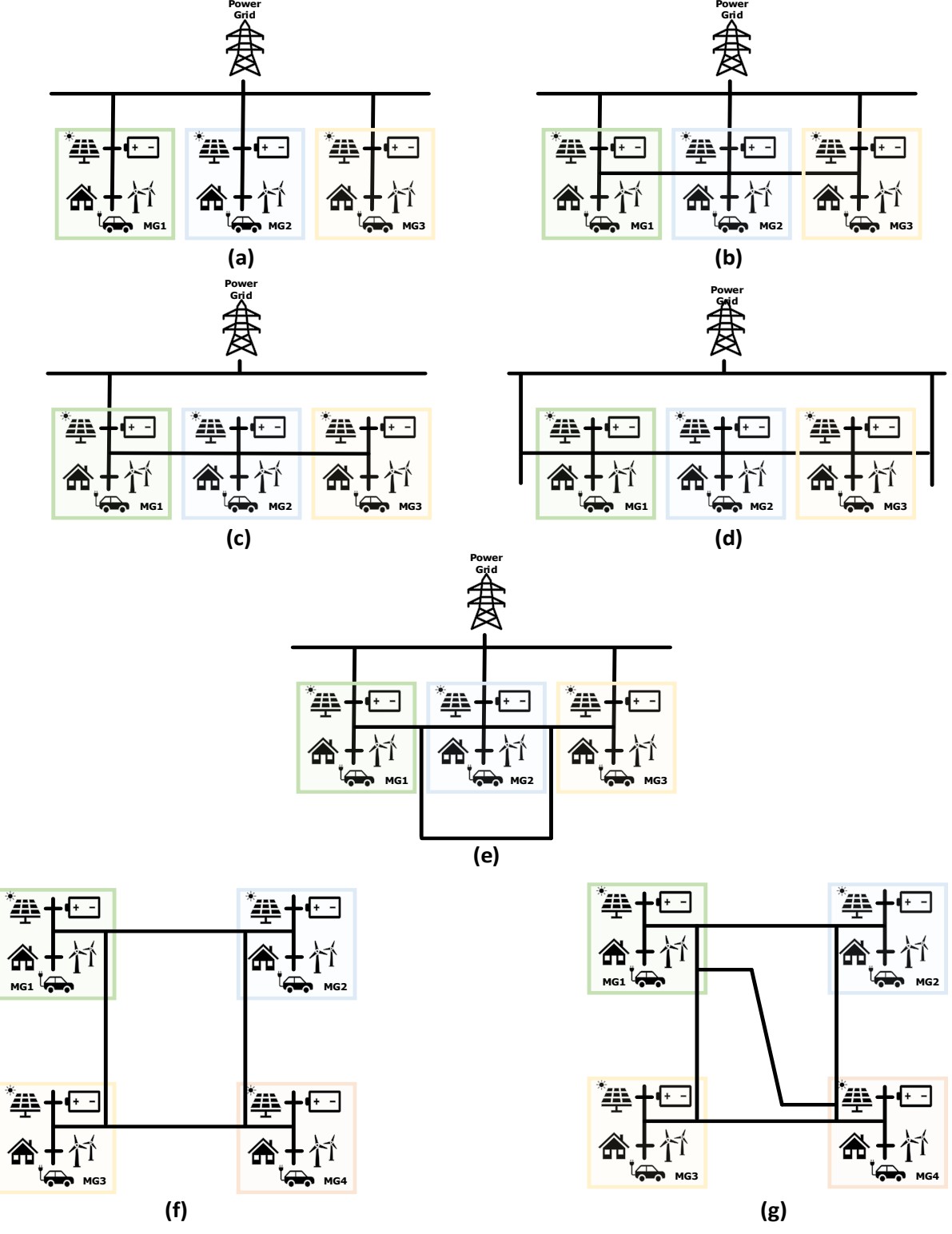

**Figure 3.** The architecture of networked microgrids: (**a**) radial topology; (**b**) daisy-chain topology; (**c**) serial microgrid on a single distribution feeder; (**d**) interconnected microgrids on multiple distribution feeders; (**e**) mesh topology; (**f**) ring formation; (**g**) grid-series-interconnected microgrids.

**Table 1.** Review of NMG architectures reported in the literature.

| Networked Microgrid Architecture | Advantages | Disadvantages | References |
|---|---|---|---|
| Radial topology | • Simple and straightforward design.<br>• Good for small microgrids. | • Single point of failure.<br>• Not as fault-tolerant as other architectures | [38,39] |
| Daisy-chain topology | • Easy to install and expand.<br>• Low cost.<br>• Good for small microgrids. | • Single point of failure.<br>• Not as fault-tolerant as other architectures | [40,41] |
| Mesh topology | • Redundancy and fault tolerance.<br>• Can withstand multiple failures.<br>• Good for large microgrids. | • Complex design and higher installation costs.<br>• Can be difficult to manage. | [42] |
| Ring formation | • Redundancy and fault tolerance.<br>• Good for medium-sized microgrids. | • More complex and expensive to install.<br>• Can be difficult to manage | [43–45] |
| Serial microgrid on a single distribution feeder | • Efficient utilization of resources.<br>• Simple and easy to install.<br>• Good for small microgrids | • Limited fault tolerance.<br>• Not as fault-tolerant as other architectures.<br>• Can have a single point of failure. | [46] |
| Interconnected microgrids on multiple distribution feeders | • Enhanced reliability and resilience through distribution feeder diversity.<br>• Can withstand multiple failures.<br>• Good for all sizes of microgrids. | • Most complex and expensive to install<br>• Can be difficult to manage. | [47] |
| Grid-series-interconnected microgrids | • Allow for redundant power supply paths and reduce the impact of single-point failures.<br>• Can improve power quality. | • Complex design and coordination.<br>• Can be expensive to install and operate.<br>• May not be suitable for all areas | [3] |

NMGs can be joined to the main grid through one or more distribution feeders. If just one of many serially linked MGs is connected to the main grid, the design is known as serial MGs on a single distribution feeder, as illustrated in Figure 3d. If the MGs are linked to the grid by more than one feeder, as in Figure 3e, the configuration changes to NMGs on multiple distribution feeder architectures [33,48]. In contrast to different grid-connected architectures, MGs can be connected away from the main utility, forming a ring architecture, as shown in Figure 3f, or with interconnections in between them, as shown in Figure 3g. Thus, this system must be robust enough to maintain constant voltage and frequency levels. This kind of topology offers better performance in terms of stability and reliability. During fault conditions, the faulted section will be disconnected from the system, while the rest of the MGs will continue to operate without any complete outages [3].

To conclude this part, there is no particular configuration for any NMG systems. They can be reconfigured in islanded or grid-connected mode to optimize their operating states during the connection or disconnection of any single MG affected by any disturbance or while supporting the entire system. In this regard, advanced control strategies are crucial for implementing an efficient energy management system, managing disruptions and fault occurrences, and coordinating different resources for optimal power sharing in NMG systems.

### 2.2. Control of NMGs

Several challenges come to the fore when dealing with the control of NMGs. Regulating the network voltage and frequency is challenging with the many system architectures and operating conditions. Add to these the determination of the optimal power distribution among DERs, the synchronization of NMGs, and the control of the SOC and the charging and discharging power of energy storage systems (ESSs). As shown in Figure 4, several control structures have been derived from a single MG's control structure to deal with these challenges in such large-scale systems. These different structures include centralized,

decentralized, distributed, and hierarchical [49]. Some of these control structures comprise three main control levels, named primary, secondary, and tertiary control.

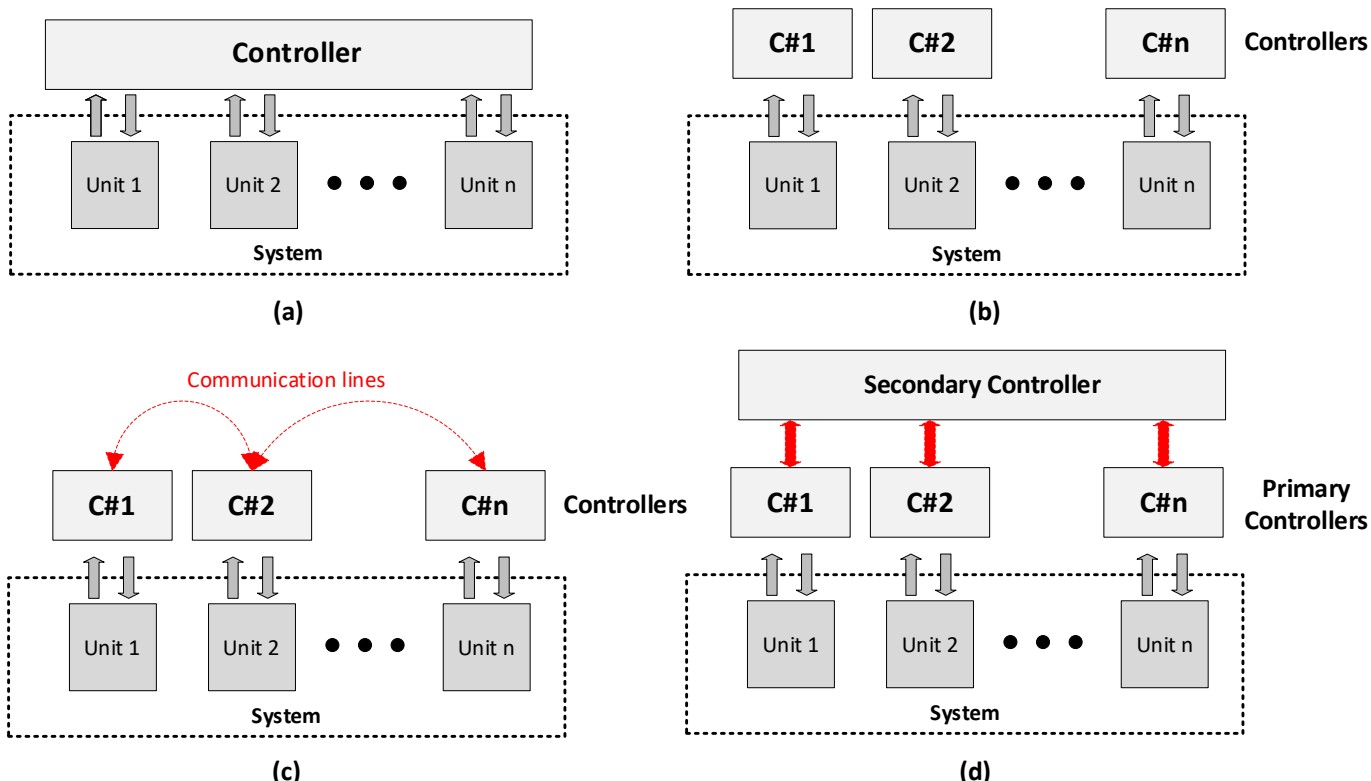

**Figure 4.** Conventional control structures: (**a**) centralized; (**b**) decentralized; (**c**) distributed; (**d**) hierarchical. In subfigure (**d**), the red arrows represent the flow of control and information between different hierarchical levels in the control structure.

Various control structures are used in networked microgrids (NMGs) to provide optimal operation and coordination across many microgrids.

(1) Centralized Control:

The primary goals of centralized control in NMGs are to minimize operating costs and maximize the overall optimization by coordinating and controlling multiple microgrids within a network. As depicted in Figure 4a, centralized control requires data gathering from essential components of each microgrid. Control and management operations in the controller are executed based on this acquired information to ensure correct and efficient functioning. While centralized control allows for easy coordination between different microgrids, it introduces the risk of a single point of failure for the entire NMG. Furthermore, the computational and communication resource requirements of these controllers limit their scalability, making them more suitable for small-scale microgrids [3,43,48].

(2) Decentralized Control:

Decentralized control in MGs refers to control mechanisms that operate independently without relying on input from other system parts. The controller utilizes only local information, as illustrated in Figure 4b, to regulate the respective unit. An example of decentralized control is droop control, which enables power sharing among distributed generation units without communication. However, the precision of decentralized control is limited by the system setup. Decentralized control enhances the autonomy and resilience of microgrids against external disruptions and attacks [4]. However, the absence of cooperation between local regulators restricts the ability to achieve globally coordinated behaviors, leading to a potential decline in NMG performance [50,51].

(3)    Distributed Control:

To overcome the limitations of centralized and decentralized control and achieve an overall optimal operation, distributed control, as depicted in Figure 4c, shares global information with neighboring microgrids and the distribution system operator (DSO). Recent advancements in communication technologies have made the implementation of distributed control in practical applications possible. These controllers require minimal bandwidth and offer a high degree of flexibility to integrate new microgrids, facilitating plug-and-play operation [30,47].

(4)    Hierarchical Control:

While NMGs are complex systems that require robust and intelligent control schemes, it is not feasible to manage the entire system using solely centralized or decentralized control structures. Hence, a hierarchical control scheme is widely accepted as a standardized solution for NMG management. This hierarchical structure consists of three essential control layers, as depicted in Figure 5. The primary control layer enables direct device control through power electronic systems, utilizing V/f, P/Q, and P/V control strategies. The microgrid controller is the secondary controller, synchronizing device controls, improving power quality, and adjusting the power flow. The tertiary control layer involves the use of Distribution Management Systems (DMSs) and Advanced Distribution Management Systems (ADMSs) to coordinate the optimal operation and load forecasting among all NMGs [52].

2.2.1. Primary Control

The fastest operating level is the primary control. The main function of this control is to interact directly with the power converters at each distributed resource within the MGs to stabilize the system under any disturbances or transients. Usually, this occurs with minimal communication between the different resources or, on many occasions, without communication [53]. To regulate the system voltage and current, converter output control usually consists of two main loops, an outer one for the voltage and an inner one for the current. The PI controller is the most common control method for both loops, which combines power performance with system dynamics. Thus, adaptive PID and nonlinear controllers based on Lyapunov theory were proposed for safe and high performance [54,55]. Droop control is widely used for power sharing between different resources, but it has limitations with nonlinear loads, instability issues, and low transient performance [56]. To address these challenges, several improvements have been made. For instance, a new complementary control loop was inserted into the conventional droop control scheme to enhance the power-sharing strategy and microgrid stability [57]. Power oscillation was dampened by combining virtual impedance and frequency droop [58], and power-sharing accuracy was improved through the compensation and tuning of droop coefficients [59,60]. A different power management technique with a frequency-based control strategy was proposed [61]. Research has also focused on decentralized power apportionment and active DC bus signaling to enhance shared power between microgrids [62,63]. These advancements demonstrate ongoing efforts to overcome the limitations of droop control and improve power-sharing strategies. Recent studies have explored innovative approaches, including fast power calculation algorithms considering nonlinear loads [64–66].

Islanding detection (ID) is yet another primary-level control concern. This mode of operation can occur due to scheduled operation or in the case of faults, and in both situations, the MG's control strategy should maintain the system's operability [52]. The detection methods can be classified into remote and local schemes. Remote schemes rely on communication between the main grid and MGs. When the link of the conveyed signal between the upstream network and the DG fails, this is referred to as a loss of main (LOM). These techniques are secure and apply to all DG types. They are, however, costly for tiny DGs [67]. Local schemes can be subdivided into active, passive, and hybrid methods and depend on measuring electrical quantities at the point of standard coupling

(PCC). A popular strategy is to assess the system frequency or voltage as a passive means of determining whether islanding is required [68] or to use disturbance injection as an active method for the same reason [69]. Lately, promising new detection techniques have been introduced, including continuous and discrete wavelet transform methods [70,71], hybrid methods combining features of active and passive techniques [72], and event index values [73], which were effectively able to differentiate between islanding and non-islanding operating modes.

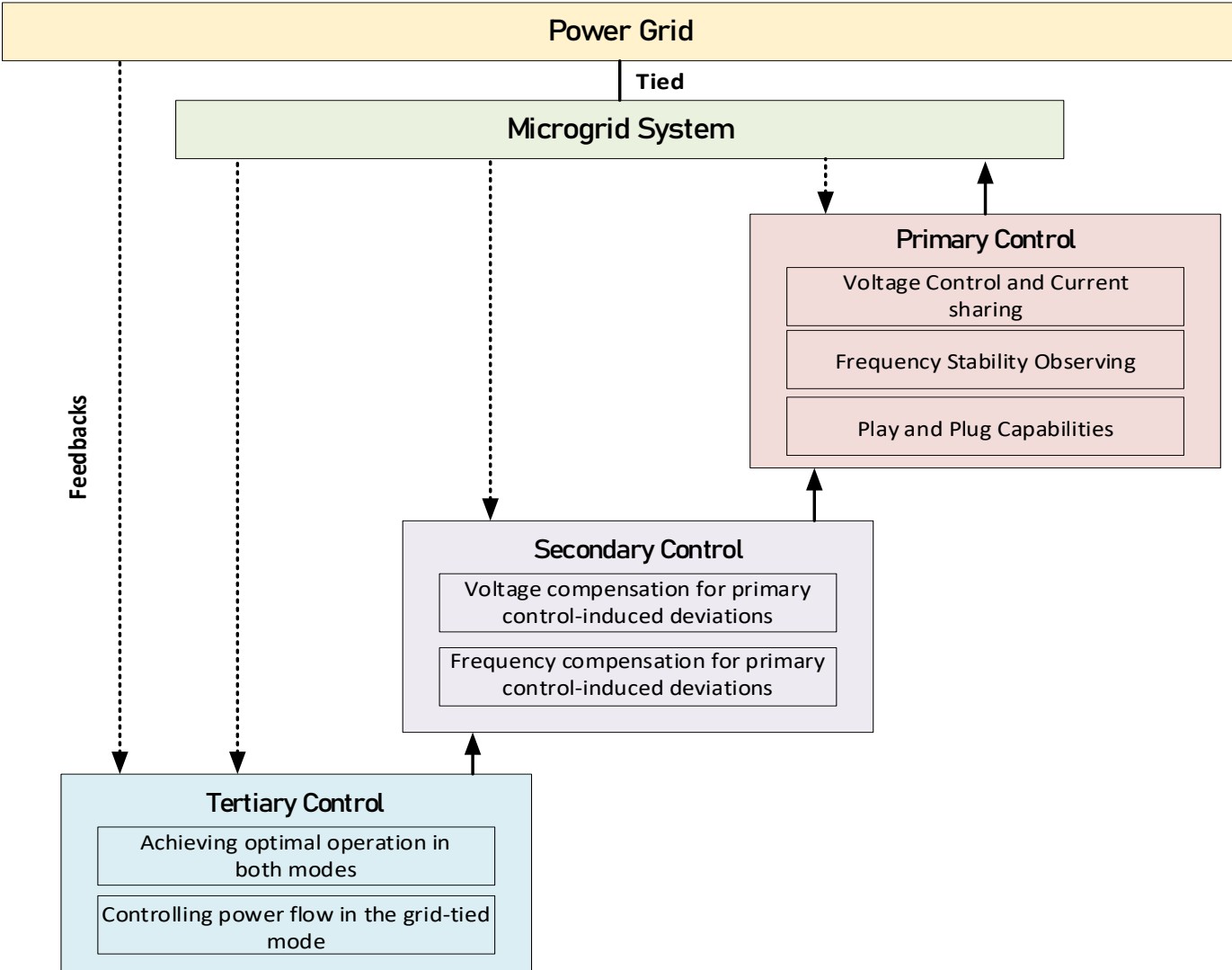

**Figure 5.** Hierarchical control structure. Hierarchical control structure. Solid arrows represent downward flow of commands and data, indicating hierarchical decision-making. Dashed arrows depict upward feedback loops for monitoring and adjusting control actions based on system feedback.

2.2.2. Secondary Control

Secondary control moderates the tertiary and primary controls, addressing the discrepancy between the tertiary level's set values and the available power in the MG measured at the primary level. In addition to correcting the voltage and frequency deviations not solved by primary control, it ensures that the upper control limits are not exceeded by sending adjusted power references to distributed energy resources (DERs) to keep the MG reliable and cost-effective while avoiding voltage and current violations. Therefore, it is also known as a microgrid energy management system (EMS) [72]. This mission is more challenging in isolated MGs with highly variable energy resources, as the unit dispatch commands

should be fast enough to overcome sudden changes in the load and non-dispatchable resources. Secondary control is the highest hierarchical level for islanded microgrid control. It operates slower than primary control, which reduces the communication bandwidth by utilizing sampled measurements of MG variables. While in grid-connected operation mode, the secondary control speed rate is between the tertiary and primary time frames [56].

Since most research efforts are dedicated to improving primary control to enhance the overall system performance, which will be reflected in a reduction in the duties performed by secondary control, there needs to be more research dealing with that part. As mentioned above, one of the main secondary control duties is regulating the voltage and frequency signals. This can be achieved by regulating the voltage of the weakest buses [74] or sensitive load buses [75]. At the same time, [76] presented a distributed-consensus-based voltage regulator to reduce the average voltage variation across the MGs. To avoid a complex control structure separated from communication networks, a proposed controller merging the primary layer's power-sharing technique with the secondary layer's voltage regulation is used to enhance the active-power-generated voltage signal [77]. A distributed secondary voltage and frequency control (SVFC) strategy for islanded microgrids was designed using multi-agent cooperative control techniques [78]. The proposed secondary control system smooths the shared active and reactive powers between DERs.

### 2.2.3. Tertiary Control

Tertiary control appears at the top of the MG hierarchical control hierarchy and has the slowest response. The primary role of tertiary control is to manage power and energy with specific goals, such as balancing energy storage, reducing power losses, and minimizing the operational cost among MGs. In contrast to secondary control, which manages power dispatch through voltage and frequency regulation and power quality, tertiary control focuses more on emphasizing electricity market participation, providing ancillary services, and administrating spinning reserves [79]. Before implementing any control scheme, three conditions should be carefully considered: the system's reliability, economics, and optimal state. Two of these three factors, namely, economics and the ideal state, are intelligently reached using the tertiary controller. Each system wishes to continue operating in an optimal or near-optimal condition. As a result, the tertiary controller employs several optimization strategies to attain the abovementioned aim. To achieve the intended optimal state of the system, the optimization process uses updated input and complicated mathematical computations. This control level often executes all these optimization operations in several minutes, sending signals to secondary-level controls at microgrids and other subsystems that comprise the whole grid [80].

There are three main optimization techniques: genetic algorithms (GAs), particle swarm optimization (PSO), and consensus algorithms [81]. In [82], a GA was utilized as a global optimization tool, including all the DG parameters and states of charge (SOCs) of energy storage systems (ESSs) during the optimization process to maximize the system efficiency. We adopted a centralized optimization strategy for power sharing based on the PSO proposed in [83], which has power loss and SoC as the same objective function. The results proved that the PSO method was the finest optimization technique, but its poor convergence in a complicated multi-model system renders it unsuitable. Hence, an upgraded version of PSO, the complete learning particle optimization approach, was employed to acquire optimal droop parameters and to execute the intended microgrid operation by resolving the problem mentioned above [84]. The consensus is an agreement between equal quantities based on an agent's status, and after executing some mathematical analyses with the help of tools such as control or matrix theories, each agent attempts to converge to one position. Consensus control strategies can efficiently ensure optimal dispatching [75], guarantee power sharing in addition to frequency and voltage stability [76], and smoothly enhance the overall system performance by sharing the harmonics and imbalances among the converters [85].

### 2.3. Protection of NMGs

The operation of networked microgrids (NMGs) with a distribution system significantly enhances the reliability and resiliency of the power supply by enabling the high penetration of locally available distributed energy resources (DERs). However, several issues in properly operating multiple MGs need to be addressed. Providing proper protection coordination is one of the most critical issues in networked MGs' (NMGs') coordinated operations [48]. One specific challenge appears when several microgrids (MGs) are integrated into the distribution feeder. This integration results in a substantial rise in the short-circuit level of the feeder. In comparison to what would be anticipated from a single MG, the combined power contributions from the MGs can cause a significant increase in fault current levels [86]. Hence, conventional protection plans for networked systems may not be appropriate since they depend solely on the short-circuit current magnitude for the unidirectional pattern of the fault currents.

According to [86], investigations demonstrated that the existing overcurrent protection technique for distribution feeders might continue to be employed with minor adaptations for the existing networks without totally redesigning the protection scheme, which is sometimes challenging in a utility environment. A practical protection coordination approach designed explicitly for networked microgrids (NMGs) was investigated in this paper. This approach utilizes numerical directional overcurrent relays (DOCRs) with single and dual settings, widely employed in power distribution systems. Using DOCRs with single and dual settings allows for precise and selective fault detection and isolation within NMGs. Single-setting DOCRs protect the primary network and individual microgrids against faults. They are set based on the rated current and characteristics of the network elements to ensure optimal protection. In [87], an innovative setting-group-based system is provided and validated to protect NMGs employing directional overcurrent relays. The suggested approach can offer adequate protection to all microgrids under all feasible interconnections between microgrids and utility grids. Using k-means clustering, a vector is constructed for each conceivable connectivity of microgrids in the system. The optimal relay settings for every set are computed using a nonlinear-programming-based technique. Protection is crucial in networked microgrids (NMGs) to ensure the system's safety, reliability, and stability. Effective protection schemes promptly detect and isolate faults, minimizing downtime and preventing cascading failures.

By implementing advanced protection coordination approaches, such as directional relays and adaptive settings, NMGs can enhance fault management, optimize system performance, and maintain uninterrupted power supply to interconnected microgrids.

### 2.4. Benefits and Challenges of NMGs

Several gains come with networking microgrids regarding resiliency, reliability, cost-effectiveness, stability, and better utilization of the available resources. One of the most critical features of NMGs is the ability to improve network resiliency. Under severe operating conditions, the network can maintain the system's operability. With the plug-and-play capability of NMGs, adding or removing resources will boost the system's resiliency, reliability, and robustness against failures and disturbances. To minimize carbon dioxide, utilizing renewable energy resources in terms of wind and photovoltaic systems will reduce the exploitation of fuel-based resources and global carbon emissions. Add to this that networking MGs enables the efficient integration of all available local DERs. From an economic perspective, clustering different resources will increase the overall generation of power. Since these resources do not have exact generation costs, the coordination between them can be cost-effective for NMGs [32,48,84].

To keep NMGs operable, several challenges should be solved. Stability, protection, and privacy are considered the main technical issues in NMGs. The high penetration of DERs and their uncertainties will impact the system's overall stability if not considered. Voltage and frequency regulation should be managed with precision. Developing robust control schemes that can utilize the fast-acting features of power electronic resources could

effectively solve these challenges. Another critical challenge is protection coordination. As the NMG's topologies change frequently, the fault current and magnitude will also change.

Moreover, the grid-connected operation mode will also affect the fault current direction, making it highly challenging for the protection system. Hence, adaptive protection schemes are mandated in such situations. With the help of communication infrastructure, coordination between different digital protection relays can provide solutions for many NMG systems [88]. However, integrating information and communication technology will increase the system's vulnerability. Cyberattacks targeting the communication layer in NMGs will significantly impact system stability. Denial of service, false data injection, and man-in-the-middle are the most common types of attacks targeting power networks. Thus, detecting and mitigating these threats is crucial for overall system stability [89].

### 2.5. Economics and Resiliency Aspects of NMGs

The interconnection of networked microgrids' economic benefits and resilience highlights the necessity for a complete evaluation that considers both factors. To appropriately measure the economic benefits, the stochastic character of unintentional islanding incidents must be incorporated into the energy management of networked microgrids. Unintentional islanding is a crucial aspect of microgrids, in which they seamlessly withdraw from the main distribution system during failures, allowing them to autonomously deliver energy to their isolated part. We may acquire an improved awareness of networked microgrids' economic advantages and resilience if we recognize and properly handle the inherent stochasticity of unintended islanding. This comprehensive method allows a complete analysis of the system's performance, allowing for informed decision making regarding deployment and operation. In [90,91], the authors propose networked microgrids to integrate variable renewable generation, improve economics, and enhance resiliency in microgrid systems. The probability of successful islanding (PSI) is introduced to quantify a microgrid's islanding capability, considering uncertainties in renewable energy resources, load variations, and power exchange. The paper demonstrates significant cost savings in electricity expenditure while maintaining high system resilience, highlighting the potential of networked microgrids in achieving efficient and resilient electricity supply. In [92], an adaptive robust optimization approach is proposed to minimize the total operating cost of networked microgrids under stochastic unintentional islanding conditions and conventional forecast errors of renewable generation and loads. The optimization strategy effectively reduces various cost components while improving the overall resilience of the power supply.

In conclusion, the economic benefits of networked microgrids are substantial, as demonstrated above. Networked microgrids provide a viable approach for attaining economic sustainability in the energy business through efficient resource sharing, load balancing, demand control, increased resilience, and new income prospects. The findings emphasize the importance of integrating multiple microgrids into a networked system to unlock these economic advantages and pave the way for a more cost-effective and resilient energy infrastructure. Networked microgrids demonstrate their potential to optimize energy utilization, reduce costs, and create new economic opportunities for consumers and grid operators.

### 2.6. Standards and Regulations of NMGs

The unique structure, operating mode (island or grid-connected), and configuration of NMGs necessitate standards and regulations to manage their operation. For instance, MGs' power quality (PQ) issues are different from those of the conventional power system, and clustering MGs together or with the main grid must follow some specifications regarding voltage, frequency, and harmonic limitations. Moreover, interconnecting DERs with the power grid through the PCC should be regulated with protection and islanding prevention standards to stabilize the overall system. In this regard, the IEEE 1547 series,

IEEE 2030 series, IEEE 519-2014, and IEC 61850 standards are examples of standards that cover these topics.

The IEEE 1547 standard series covers the interconnection between DERs and the electric grid. IEEE 1547 has aided in modernizing our electric power infrastructure by laying the groundwork for integrating clean renewable energy technologies and other distributed generation and energy storage technologies. IEEE 1547 specifies necessary functional technical standards and specifications, flexibility, and options for complying with equipment and operational details. In December 2013, an IEEE SCC21 workshop was convened to revise and update IEEE 1547 and tackle interoperability and interface issues between the grid and DERs, which concluded these updates in the IEEE 2030 standard. Table 2 highlights some main points introduced in these two standards [93,94]. Table 3 provides an overview of common standards related to power quality requirements, including descriptions of each standard and their specific focus areas. Table 4 presents the standard limits for current and voltage harmonics according to IEEE and IEC standards, indicating the harmonic order, limits, and total harmonic distortion (THD) requirements for different voltage levels.

**Table 2.** Topics covered in IEEE 1547/IEEE 2030 standards.

| Standard | Description |
| --- | --- |
| IEEE 1547.1 (2005) | Details the test methods for the equipment that connects the DERs to the power grid. |
| IEEE 1547.2 (2008) | Includes guidance for the use of IEEE standards during interconnection. |
| IEEE 1547.3 (2007) | Creates guidelines for inspection, exchange of information, and DER control with an electrical network. |
| IEEE 1547.4 (2011) | Provides design, operation, and integration guidelines for DER island systems with utility networks. |
| IEEE 1547.6 (2011) | Defines the standards for connecting the power system's secondary network to DREs. |
| IEEE 1547.7 (2013) | Outlines the operational actions required to assess the impact of DERs on the power system. |
| IEEE 1547.8 (2014) | Suggests identifying and expanding new design and operational processes to fully exploit DERs and power systems. |
| IEEE 2030.1 | Provides guidance for electric-powered transportation infrastructure. |
| IEEE 2030.2 (2015) | Focuses on the integration of HESS associated with electric power infrastructure, in addition to end-use applications and loads. |
| IEEE 2030.3 (2015) | Specifies the test technique for electric energy storage systems used in power systems. |
| IEEE 2030.6 (2016) | Describes a framework for monitoring the effects and evaluating the comprehensive benefits of demand response programs. |
| IEEE 2030.7 (2017) | Provides technical specifications and requirements for microgrid controllers. |
| IEEE 2030.8 (2018) | Defines testing standards for MG controllers for energy management. |
| IEEE 2030.9 (2018) | Offers guidance for microgrid planning and designing. |
| IEEE 2030.10 | Highlights the DC MGs' design, operation, and maintenance for urban and rural applications. |

**Table 3.** Common standards related to power quality requirements.

| Standard | Description |
| --- | --- |
| IEEE 519-92 | Describes the harmonic specifications and conditions of the power system |
| IEEE 1159-95 | Specifies the power quality requirements for power systems |

**Table 3.** *Cont.*

| Standard | Description |
|---|---|
| IEEE 1100-99 | Suggests the needed specifications for grounding and powering sensitive electronic devices |
| IEC 61000-2-2 | Provides compatibility rates in industrial plants for low-frequency conducted disturbances |
| IEC 61000-3-2 | Lists limitations of harmonic current emissions |
| IEC 61000-4-15 | Deals with flicker and fluctuation specifications |
| IEC 50160 | Defines distribution systems' voltage specifications |

**Table 4.** Standard current and voltage harmonics according to IEEE/IEC standards.

| Current Harmonics | | | |
|---|---|---|---|
| **Standards** | **Harmonic Order** | **Limit** | **THD** |
| IEEE 1547 | $3 \leq h < 33$ (odd) | $< (4\% - 0.3\%)$ | $< 5\%$ |
| | $2 \leq h \leq 32$ (even) | $< (1\% - 05\%)$ | |
| IEC 61000-3-2 | $3 \leq h \leq 39$ (odd) | $< (0.3\% - 0.6\%)$ | $< 5\%$ |
| | $8 \leq h \leq 40$ (even) | $< (0.2\% - 1.6\%)$ | |
| **Voltage Harmonics** | | | |
| **Standards** | **Voltage Level (kV)** | **Harmonic Limit** | **THD** |
| IEC | $V > 161$ | 1% | 1.5% |
| | $69 \leq V \leq 161$ | 1.5% | 2.5% |
| | $2.3 \leq V \leq 69$ | 3% | 5% |
| IEEE 519 | $V > 161$ | 1% | 1.5% |
| | $69 \leq V \leq 161$ | 1.5% | 2.5% |
| | $1 \leq V \leq 69$ | 3% | 5% |
| | $V \leq 1$ | 5% | 8% |

## 3. Networked Microgrid Communications

This section explains the general requirements and challenges for communication systems in an intelligent grid and investigates the impact of the communication structure on networked microgrid systems. In addition, communication technologies and standards are defined.

### 3.1. Communication Requirements for Smart Grid Systems

Communication systems ensure dependable, efficient, and secure power generation, transmission, and distribution. They facilitate the exchange of information between distributed sensing equipment, monitoring, and data management systems [95,96]. In a microgrid system, communication is frequently required in both directions between the controller and the monitored or controlled devices, and despite advanced metering infrastructure (AMI), which only offers one-way monitoring, distributed energy resources (DERs) such as solar panels and battery storage exchange information with the microgrid controller (MC). According to a report from the U.S. Department of Energy [97], distribution systems utilizing these technologies need between 9.6 and 56.0 kbps. This bandwidth can be sufficient to build reliable communication between the controller and devices and allow for real-time data exchange in the grid. Here is a list of the requirements that a smart grid's communication network must fulfill:

- Latency: Latency can be described as the time in which data move between two points in a communication network inside a smart grid. The capacity of a smart grid to successfully control and manage the flow of energy is impacted by latency, which is a crucial component of the smart grid's operation. Low latency is necessary for real-time applications such as demand response, grid monitoring, and power system safety because it helps the grid run effectively and consistently.
- Reliability: Communication reliability is the capacity of a smart grid's communication system to send data precisely and reliably. As it guarantees the proper operation of the grid's different elements, including distribution systems, renewable energy sources, and energy storage systems, it is a critical component of the smart grid. Reliable communication is provided in the smart grid by using redundant communication channels, algorithms for error detection and correction, and routine testing and maintenance of the communication network.
- Bandwidth: The smart grid communication network's bandwidth requirements must be determined since they directly impact the choice of transmission media (fiber optics, radio waves, and coaxial cables) and communication technology (e.g., 3G, LTE, and WiMAX). It is crucial to remember that if suitable precautions are not followed, the communication system's numerous endpoints could result in unmanageable bandwidth requirements [98].
- QoS: The ability of the communication network to transmit the required information with the desired degree of dependability, performance, and security is ensured by the quality of service (QoS), which is a crucial requirement for smart grid communication. Data transmission must be quick, dependable, secure, and consistent across the smart grid communication network [99].
- Scalability: Scalability is the capacity of a network or system to change to meet growing demand and to increase its capacity as necessary. Scalability is a crucial requirement for smart grid communication because the network must manage an increasing number of connected devices, an increase in data volume, and technological advances. Since the needs of the smart grid system are constantly changing, this requires a flexible and straightforward upgradable communication network [100].
- Interoperability: Interoperability is necessary for smart grid communication for different gadgets, systems, and applications to collaborate efficiently. The communication system of the smart grid should be able to link to preexisting legacy systems and newer systems and technologies without much alteration. Interoperability is critical to ensuring that different gadgets, systems, and applications can interact and transfer information, allowing for more advanced features, such as surveillance, control, and real-time data analysis [101]. To achieve interoperability, open standards and protocols such as IEC 61850 and IEC 60870-5-104 must be used to ensure that communication systems are created with modular and adaptable structures.
- Security: Security is a critical component of the smart grid's communication infrastructure since it defends the sensitive data acquired from various elements from both physical and virtual threats. Most SG apps place a high premium on ensuring end-to-end security [102]. Security measures must be immediately integrated into the communication network rather than added as an afterthought.
- Standardization is crucial to the smart grid communication system since it ensures interconnection and compatibility across different components and systems. Communication protocols, technologies, and interfaces must be standardized to facilitate the smooth integration of various components and systems, enabling efficient and effective communication.

Efficient communication is necessary for a networked microgrid system to run correctly and in coordination. In such a system, various microgrids are linked to form a more extensive network. Therefore, communication is needed to transfer data between these microgrids to harmonize the energy flow and ensure a secure and adequate power supply. The communication infrastructure used in networked microgrid systems

usually comprises wireless networks, power line communication (PLC), and cellular networks. These technologies provide the real-time observation and management of energy generation and utilization and the potential to coordinate energy exchange between microgrids to balance energy supply and demand. For instance, if one microgrid has excess energy, it can communicate with another microgrid with excess energy, communicate with another microgrid with extra energy, communicate with another microgrid with a shortage, and transfer energy to meet the latter's needs. In addition, communication also facilitates the integration of renewable energy sources, such as solar and wind, into the networked microgrid [103].

### 3.2. Smart Grid Communication Technologies

"Smart Grid Communication Technologies" refers to the communication systems and protocols used for the efficient and accurate administration, control, and monitoring of the smart grid. These technologies enable data transmission through various smart grid components, including electricity generation, distribution networks, and energy management systems. The communication technologies used in the smart grid must comply with several standards, including real-time performance, reliability, security, and interoperability. Communication in a microgrid can be wired or wireless, and several systems require a combination. Choosing the appropriate communication technology for a given situation depends on several factors, including regional characteristics, operational and technical requirements, and financial constraints [104]. The cost, ease of installation, and interference influence the decision between wired and wireless communication in microgrid environments. Both types of communication can be helpful in various situations, but what works well in one may not work well in another. Although wired connections are less susceptible to interference problems than wireless connections, they can be more expensive to establish in a complex system. On the other hand, wireless communication may be more straightforward to implement.

#### 3.2.1. Wired Communication

- Power Line Communication (PLC)

Power line communication (PLC) is a data transmission technology that utilizes the electrical grid. High-frequency signals ranging from a few kilohertz to tens of megahertz are transmitted via lines in PLC. PLC systems are classified according to the frequency band in which they operate, such as ultra-narrowband (UNB-PLC), narrowband (NB-PLC), and broadband (BB-PLC). UNB-PLC operates in the 125 Hz to 3 kHz frequency range, NB-PLC operates in the 3–500 kHz frequency range, and BB-PLC operates in the 1.8–100 MHz frequency range [105]. NB-PLC is chosen in smart grid applications, where reliability, range, and durability are the key considerations.

In contrast, BB-PLC is employed in home and building area network internet access applications with a constrained coverage area [106]. PLC technology has many uses in the smart grid, including advanced metering infrastructure (AMI), demand response, and household energy management systems. Meter readings from residences and businesses can be sent to the utility company using AMI and PLC technology. Demand response systems employ PLC technology to link with demand response equipment installed in homes and businesses to control energy use during peak demand. Home energy management systems use PLC technology to communicate with and regulate intelligent equipment, such as smart plugs, lighting controls, and thermostats [107].

Notwithstanding its advantages, PLC technology has several disadvantages. The quality of the communication signal can be impacted by electrical noise and interference from other electrical appliances, which is one of the main difficulties. Signals used for long-distance communication can also deteriorate, rendering them unsuitable for massive communication networks.

- Ethernet

Ethernet is a local area network (LAN) communication technology that employs a physical wire and a set of pre-established protocols to transport data between two devices. It was developed in the 1970s and is one of the most used communication technologies worldwide. It has a coverage range of 1 to 100 m, operates in the unlicensed 2.4 to 835 GHz band, and provides a 721 Kbps data transmission rate. The Open Systems Interconnection (OSI) architecture inspired the seven-layer communication structure used by Bluetooth-enabled devices, allowing for direct communication between two devices and communication among several devices. It provides a lower level of security than other technologies because of its potential for interfering with IEEE 802.11 wireless LAN networks and susceptibility to disruptions in the environment's communication capacity [108].

- Optical Fiber Communication

Due to its many benefits, optical fiber communication has become the leading technology for sustaining electrical power network communication. Some of these are a high bandwidth capacity, less signal deterioration, immunity to electromagnetic interference, and improved security [94]. Fiber optic communication is the best option for control and monitoring needs and backbone communication in wide-area networks (WANs). It supports high-speed data transmission over vast distances thanks to its high data transfer rates, ranging from 5 Gbps to 40 Gbps. Even though the startup investment and maintenance expenses could be high, its performance eventually makes it the ideal smart grid option [109].

- Serial Communication

Serial communication sends data via a network or computer bus one bit at a time and consecutively. It is frequently used for interfacing microcontrollers, industrial automation systems, and computer peripherals. A start bit and a stop bit are used to separate each data word in asynchronous serial communication, which synchronizes data transfer with the assistance of a clock signal. Popular serial communication technologies used for various applications include RS-232, RS-485, and USB [110].

3.2.2. Wireless Communication

- Cellular Communications

Voice, data, and multimedia content are sent and received using a network of cell towers and base stations in cellular communications. A smart grid may communicate with its many parts and equipment using cellular networks, enabling real-time power grid monitoring, control, and data exchange between devices [111]. The extensive preexisting infrastructure that makes it easy to communicate between various parts and devices and the high data transmission speed that enables quick and dependable communication are just two of the benefits that cellular networks offer for smart grid connectivity. The many cellular connection technologies now in use—including GSM, 2G, 3G, 4G, 5G, and LTE-M—offer varying coverage and data transfer speeds [112]. Nevertheless, a significant drawback of cellular networks for smart grid communication is that they are not exclusively dedicated to this purpose and are shared with other users, which might cause issues during crises.

- Zigbee

The IEEE802.15.4 standard is the foundation of ZigBee technology [113]. With two-way wireless data transmission operating at 2.4 GHz and 868 and 928 MHz, IEEE802.15.4 is a cost-effective, high-efficiency, low-rate standard for personal area and peer-to-peer networks. Based on the definitions of the physical layer (PHY) and media access layer (MAC) in the IEEE802.15.4 standard, the ZigBee Alliance expands the network layer (NWK) and application layer structures (APL) [114]. Micro-power wireless communication systems, including ZigBee, are described in Table 5, along with some of their typical characteristics.

**Table 5.** Overview of wired/wireless communication technologies in smart grid.

| Comm Technology | Type | Data Rate | Range | Security | Cost | Applications | Advantages | Disadvantages |
|---|---|---|---|---|---|---|---|---|
| PLC | Wired | 2–500 Mbps | Varies | High | Low/moderate | Smart grid, home automation, building automation | Uses existing electrical wiring, reducing installation costs. | Performance can be affected by noise and signal attenuation in power lines. |
| Ethernet | Wired | 10 Mbps–100 Gbps | 100 m | High | Low/moderate | Industrial control, data centers, offices | High data rates and low latency. | Limited range compared to wireless technologies. |
| Fiber optic | Wired | 5–40 Gbps | 40 km/ up to 10 km | High | Expensive | Data centers, high-speed internet, long-distance communication | Very high data rates and long-distance capabilities. | Expensive installation and equipment costs. |
| Serial Com | Wired | 110 bps–4 Mbps | 15 m | Low | Low | Industrial control, automation, IoT | Simple and cost-effective for short-range communication | Limited data rates and range. |
| Cellular Com | Wireless | 5 Gbps and beyond | Long distance | High | Expensive | Smart grid, distribution automation, IoT | Wide coverage and long-distance capabilities. | Relatively high cost and power consumption compared to other wireless technologies. |
| Zigbee | Wireless | 20–250 kbps | 10–100 m | Low | Low | Smart homes, industrial automation, IoT | Low power consumption. | Limited data rates and range. |
| Wi-Fi | Wireless | 54 Mbps–10 Gbps | 100 m | Medium | Moderate/expensive | Home and office networks, public hotspots, IoT devices | High data rates and wide availability. | Limited range compared to some other wireless technologies. |
| LoRaWAN | Wireless | 0.3–50 kbps | 2–5 km (urban)/15 km (rural) | Medium to High | Low/moderate | Smart agriculture, smart cities, smart homes, IoT devices | Long-range communication capabilities. Low power consumption. | Lower data rates, relatively high latency. |

- Wi-Fi

Wi-Fi is a wireless communication technology that provides high-speed internet and network connectivity using radio waves. It transmits data using the IEEE 802.11 standards and operates in the 2.4 and 5 GHz frequency ranges [115]. Wi-Fi has become a widely used technology. Multiple devices may connect to a single Wi-Fi access point, making it an ideal choice for households and small enterprises. However, interference from other signals in the 2.4 GHz frequency spectrum, such as Bluetooth devices, might cause interference, and the technology may have a limited range and penetration through walls. Wi-Fi network security is also an issue because data transferred over the airways might be intercepted by unauthorized users [116]. Despite these restrictions, Wi-Fi remains a popular wireless communication technology, particularly in the consumer and small business industries.

- LoRaWAN (Long-Range Wide-Area Network)

LoRaWAN (Long-Range Wide-Area Network) is a wireless communication technology utilized in the Internet of Things (IoT) and smart grid applications. It is a low-power, long-range technology that operates in the sub-GHz spectrum, making it suitable for long-distance, low-data-rate communication with little power consumption [117]. By enabling wireless communication between intelligent devices without laborious local setups, the LoRaWAN standard streamlines the adoption of the Internet of Things. It grants more freedom to businesses, customers, and innovators [118].

Several strategies can be used to overcome the disadvantages of different types of communication. Data reliability is increased for PLC by integrating noise filtering, signal conditioning, and error-correcting codes. Ethernet can be extended and become more mobile by using network switches and repeaters, while fiber optic lines provide faster data speeds. Media converters make it possible to use the current infrastructure.

Signal boosters, error-checking protocols, and sophisticated serial communication protocols improve serial communication performance. In cellular communication, selecting trusted service providers, improving antenna positioning, and employing signal amplifiers can increase coverage. Zigbee networks can be used to overcome restrictions by installing more devices, strategically placing routers and repeaters, and adopting techniques such as frequency hopping. Wi-Fi range extenders can increase coverage, encryption algorithms can improve security, and channel selection and network design can improve performance. LoRaWAN may extend network coverage with additional gateways and adaptive data rate settings, while redundancy and failover techniques boost dependability. Applying these solutions to specific requirements can overcome shortcomings and improve overall communication system performance.

### 3.3. Impact of Communication on Networked Microgrid Systems

Communication is crucial for coordination and collaboration between microgrids in a networked microgrid system. Wireless and wired communication technologies are used to facilitate the exchange of information between microgrids, the central energy management system, and end users. Data on weather conditions, energy production and consumption levels, outages, and system failures are transmitted in real time to enable more efficient management and optimal power supply planning.

Wireless networks, such as Wi-Fi networks, wireless sensor networks (WSNs), and short-range communication (NFC) networks, are often used to connect renewable energy generation equipment, such as solar panels and wind turbines, to microgrids. These technologies are also used for real-time data collection, power generation monitoring and control, load management, and power supply planning.

Wired networks such as Ethernet, fiber optic, and powerline communication (PLC) connect microgrids to the central energy management system and end users. These technologies enable faster and more reliable data transmission and more secure and private communication. One of the most significant impacts of the communication layer on NMG systems is system reliability. Microgrids can only exchange power and data efficiently

with reliable communication. This can result in communication delays, packet losses, and other problems, resulting in a loss of synchronization between the microgrids. As a result, the NMG system may have an unbalanced load distribution, with certain microgrids overloaded and others underused. This unbalanced load distribution can lead to power quality concerns, such as voltage and frequency fluctuations, which can cause the NMG system to become unstable. In some situations, it may also lead individual microgrids to function in islanded mode, separated from the broader power grid. This can result in power outages, harming critical infrastructures such as hospitals, data centers, and other vital services.

In networked microgrid systems, the security of the communication layer is a critical factor that must be evaluated. Cyberattacks on NMG communication networks can represent a significant risk to the system. Hackers can use communication network weaknesses to obtain unauthorized access to the system. Infiltration of this type might result in data theft, manipulation, or even system blackouts. Because the communication layer permits the interchange of sensitive data and control signals across microgrids, its security is critical. Furthermore, a cyberattack on a networked microgrid might result in substantial economic losses. Fixing and repairing the system can be expensive, and additional expenses related to consumer compensation and legal processes may be incurred.

The communication layer of a networked microgrid (NMG) system as shown in Figure 6 is critical to ensuring the appropriate functioning and management of the complete system. The communication network is in charge of sending control signals, instructions, and feedback across the many microgrids, allowing them to collaborate in a coordinated and efficient manner. However, if communication fails or the network becomes overloaded, it might result in a loss of control and synchronization between microgrids, making the system unstable. For example, if there is a delay or even a loss of control signals because of network congestion, the microgrids will not be able to receive the necessary commands in time, and therefore, the power output will not match the demand. As a result, the NMG system's overall dependability and stability will be at risk.

Furthermore, the communication layer can impact the performance of the networked microgrid system very negatively since NMG communication networks are frequently complex and require significant bandwidth to send data for real-time management. Low bandwidth or mediocre quality of service (QoS) can cause severe damage, such as packet losses or reduced system performance. To reduce energy use and increase efficiency, energy management systems (NMGs) are increasingly being employed in buildings. A crucial component of these systems is the communication layer, which enables the linking of the various devices and parts of the system and the transmission of the data required for decision making.

Redundancy and fault tolerance must be considered while designing the communication layer to guarantee that the system will function effectively during a communication breakdown. Backup communication links, additional routing routes, and innovative technologies such as LoRaWAN, ZigBee, and Wi-Fi can accomplish this.

### 3.4. Communication Protocols and Standards

Communication protocols are essential to a networked microgrid system's effective and reliable operation. Devices and systems can communicate and share information because of sets of rules and standards called communication protocols. Various criteria, such as functionalities, implementation, and use cases, can classify them. These protocols allow information to be exchanged between software applications running on multiple devices.

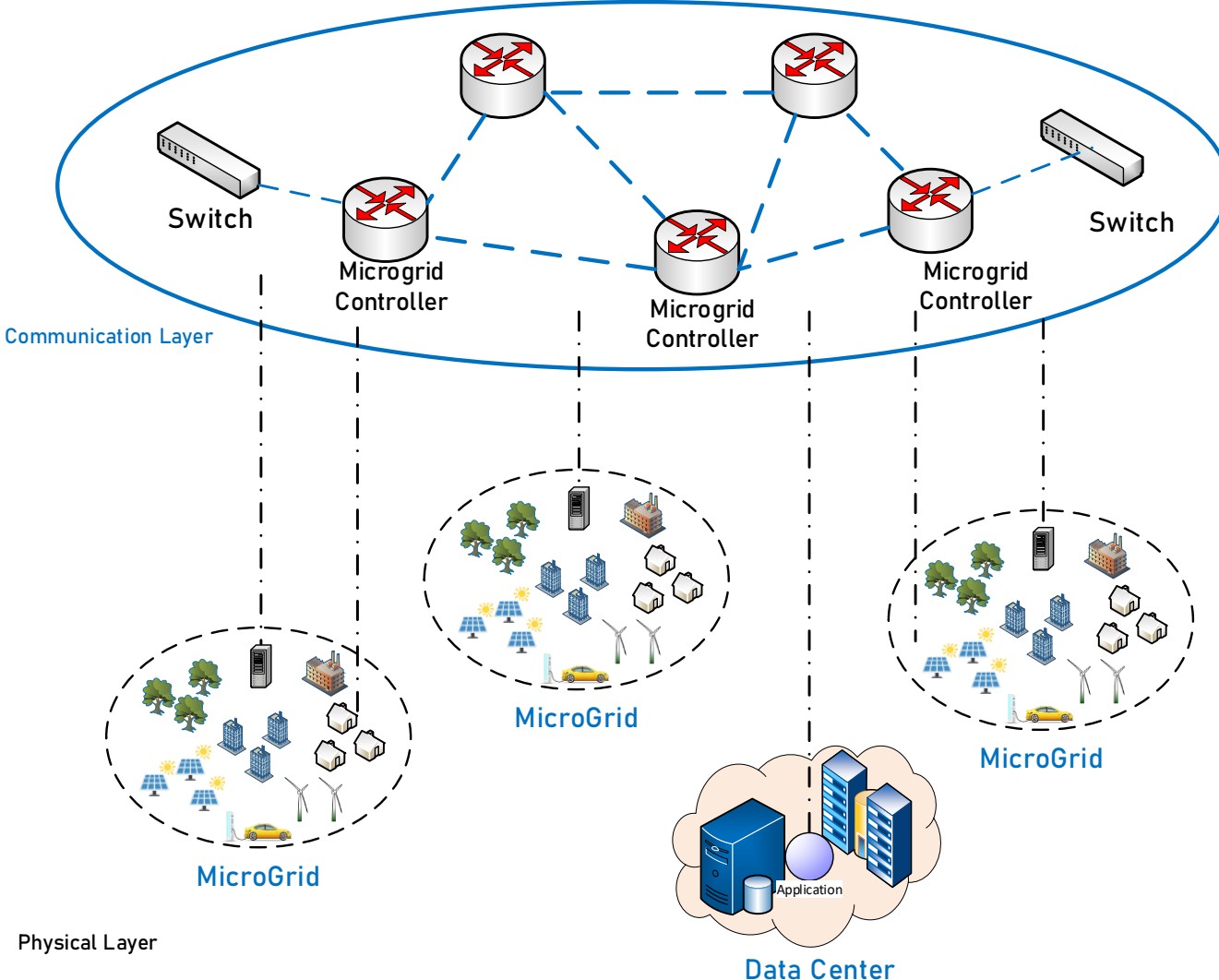

**Figure 6.** Overview of NMG communication network framework.

- IEC 61850 IEC 61850 was initially introduced in 2003 to integrate various components within a grid, such as protection devices, sensors, and control systems. It aims to enhance interoperability and flexibility by providing a standardized interface between various devices and systems [119–121]. The IEC 61850 standard is divided into several parts, each defining a specific protocol aspect. These include:

  - System Aspects (IEC 61850-1, IEC 61850-2, IEC 61850-3, IEC 61850-4, and IEC 61850-5): These parts outline the general and particular subjects and specifications for communications in a substation. They cover issues such as device information sharing and substation topology in addition to the communication network.
  - Configuration (IEC 61850-6): Based on the XML schema, this part describes configuring Intelligent Electronic Devices (IEDs) compatible with IEC 61850. The SCL offers a standardized method for defining a substation's logical and physical elements and the communication links that connect them.
  - ACSI (Abstract Communication Service Interface): This is a crucial part of the IEC 61850 standard in power grid automation systems. This interface is split into four sections, each with a specific communication and data-handling function.

1.  IEC 61850-7-1: Specifies the fundamental models of information that the system utilizes, including information on switching, status, and measurement data.
2.  IEC 61850-7-2: Specifies the abstract services utilized in the system to manipulate and manage data, enabling compatibility across heterogeneous hardware and software.
3.  IEC 61850-7-3: Outlines the typical data classes utilized inside the systems, including data types and communication services.
4.  IEC 61850-7-4: Describes the concept of logical nodes, which are data object abstractions used to describe the functions and data of a system uniformly.

- Mapping sections (IEC 61850-8 and IEC 61850-9) explain how information is mapped and exchanged between systems using one of the mapping methods (protocol stacks) outlined in the IEC 61850 standard.
- Testing (IEC 61850-10): This document specifies a testing procedure to ensure that gadgets adhere to the IEC 61850 standard. The ability of devices from various manufacturers to function together seamlessly depends on this.

IEC 61850 can be used in networked microgrid systems to help communicate and coordinate different microgrids and energy management systems dispersed across diverse sites. By establishing a single communication platform, the standard can facilitate the exchange of control signals, monitoring data, and system status data between different microgrids and energy management systems. This allows other microgrids to work together in a more efficient and coordinated way, which can improve the stability of the power grid, reduce costs, and improve the reliability of the energy supply.

- DNP3

The DNP3 communication protocol was first presented by Westronic Inc. in 1990 and made available to the public in 1993. Since then, the protocol has gained wide acceptance for building robust, open, and efficient SCADA systems [122]. In critical infrastructure settings, DNP3 is a reliable and efficient communication protocol. The master or server at the control center can more easily receive measurement data from an outstation or client in the field [123]. DNP3 is used in microgrid systems to monitor and manage distributed energy resources, including solar panels, wind turbines, and energy storage systems. The protocol enables communication between the microgrid control system and the various microgrid components. DNP3 supports time synchronization, which is critical for microgrid systems dependent on distributed energy resources. It enables the control system to manage energy flows and balance supply and demand in real time by providing accurate and synchronized time stamps.

- Modbus

The Modbus protocol was initially developed by Modicon in 1979 as a messaging framework for facilitating communication between intelligent devices that function as master–slave systems [124,125]. Since then, it has become a standard communication protocol for many types of industrial equipment and sensors. The Modbus protocol was initially developed for asynchronous serial lines such as RS-232 and RS-485, which connect intelligent devices. RTU and ASCII transmission modes are supported, but only the former is required. As shown in Figure 7 for RTU mode, a Modbus frame for serial lines consists of a single Modbus Protocol Data Unit (PDU) inside a Modbus Application Data Unit (ADU) as shown in Figure 8.

Modbus can improve communication between parts such as inverters, generators, and energy storage systems in standalone microgrids. These components can communicate status and performance data using Modbus, which enables the microgrid controller to control energy flows and balance supply and demand in real time. This may contribute to raising the microgrid's overall effectiveness and dependability. In addition to that, Modbus is a dependable and widely adopted protocol that can be used to facilitate communication and data exchange in networked microgrid systems.

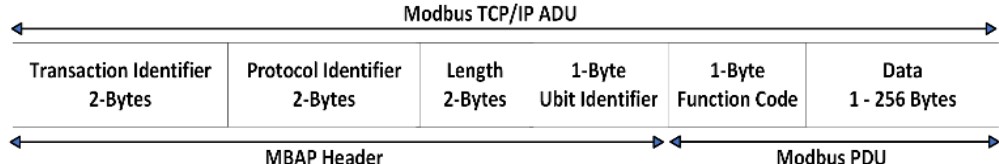

**Figure 7.** Modbus frame construction for serial line transmissions in RTU mode.

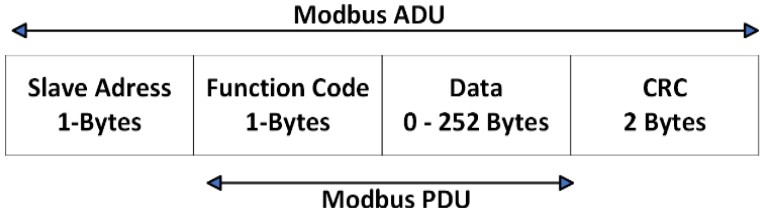

**Figure 8.** Modbus TCP frame.

- OPC UA

OPC UA is a set of standard protocols that enable interoperability between automation and control applications, field systems and devices, and enterprise applications in the process control industry, providing a communication infrastructure and information model standardized as IEC 62541 by the OPC Foundation [126]. Figure 9 depicts the two backbones, the transport model, and the data model, in the architecture of OPC UA.

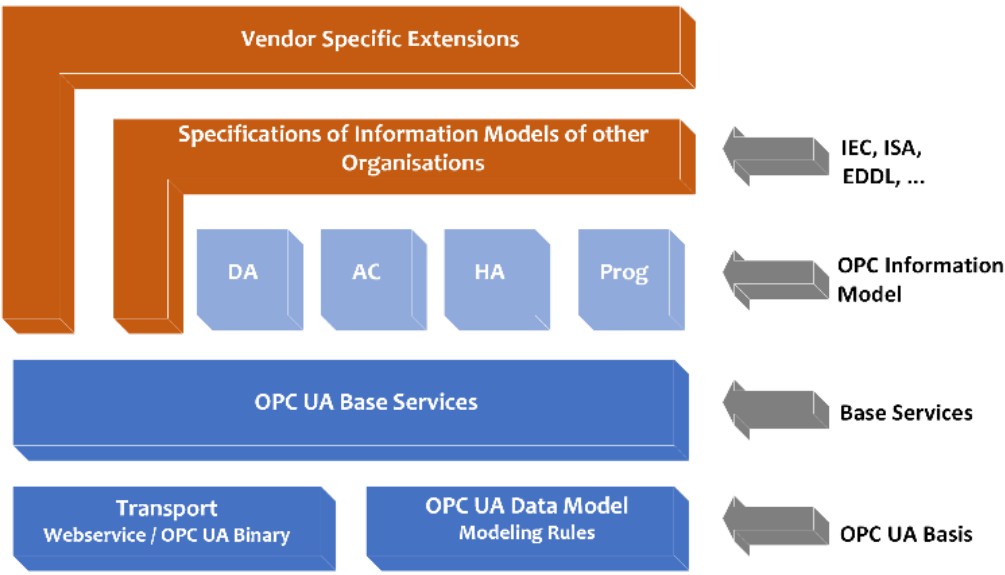

**Figure 9.** OCP UA architecture.

In a networked microgrid system, OPC UA facilitates data interchange and communication across components, assuring effective and coordinated control and offering innovative cybersecurity features such as encryption and authentication.

- MQTT

MQTT (Message Queuing Telemetry Transport) was developed in 1999 by Andy Stanford-Clark of IBM and Arlen Nipper of Arcom Control Systems. It is a lightweight messaging protocol that uses the publish/subscribe principle and operates on TCP/IP. A client publishes messages to a broker, which can be subscribed to by other clients and stored as future subscriptions. Clients can subscribe to multiple topics to receive all published messages, and each message is sent to a topic address [127]. Figure 10 illustrates the elements and process of MQTT protocol.

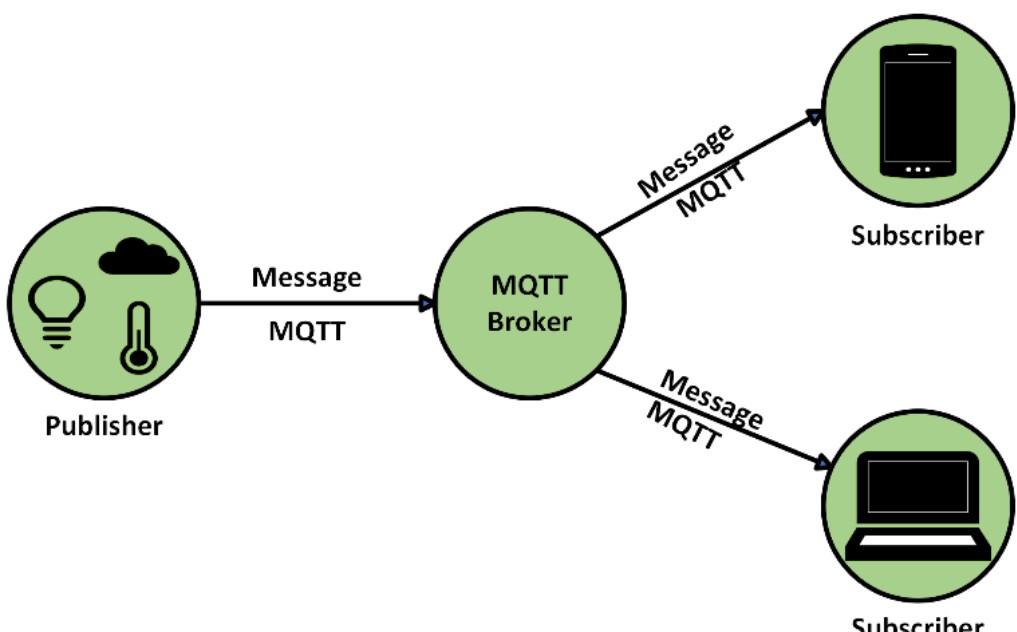

**Figure 10.** MQTT publisher/subscriber architecture.

MQTT works best for sizable embedded system networks that must be supervised or managed from an online back-end server. Device-to-device communication and multicasting data to numerous receivers are not intended uses. It is a straightforward messaging protocol with a few control options.

- AMQP

Like MQTT, AMQP operates based on the principle of publish/subscribe. AMQP is an open and standardized application layer protocol built for messaging settings, emphasizing Internet of Things (IoT) applications. It facilitates reliable message exchange between devices and systems, even in dispersed and diverse situations [128]. It provides messaging-related capabilities such as dependable queuing, flexible routing, and transaction support. It also supports topic-based publish-and-subscribe messaging, allowing efficient communication and data sharing in message-oriented contexts [129].

- CoAP

CoAP uses the client/server communication pattern, where a client sends a request to a server with a method code and URI for a resource, and upon processing the request, the server sends a response back to the client with the information requested [130]. CoAP can be used in networked microgrids to exchange data between resource-constrained IoT devices that require lightweight communication protocols. CoAP's client/server communication pattern enables efficient communication between devices and servers in a microgrid, allowing them to communicate information on the state of the grid, energy demand, and renewable energy source availability. The controllers and management systems may utilize these data to improve the functioning of the microgrid, delivering a reliable and efficient energy supply to users. Furthermore, Uniform Resource Identifiers (URIs) via CoAP enable simple identification and access to specified resources in the microgrid network, facilitating device discovery and administration.

- BACnet

BACnet is a vendor-independent communications protocol for Building Automation and Control Networks (Figure 11). BACnet specifies a set of rules that govern how devices should communicate effectively. Because of their normalization, BACnet devices can communicate with one another regardless of manufacturer. BACnet is a four-layer protocol stack that is more than just an application layer protocol. Different protocols can be used in

the physical and data link layers to accommodate different environments. The network layer enables the connectivity of two or even more BACnet networks. The application layer is in charge of actual data exchange between BACnet devices, BACnet objects, properties, and services play an important role in the application layer [131].

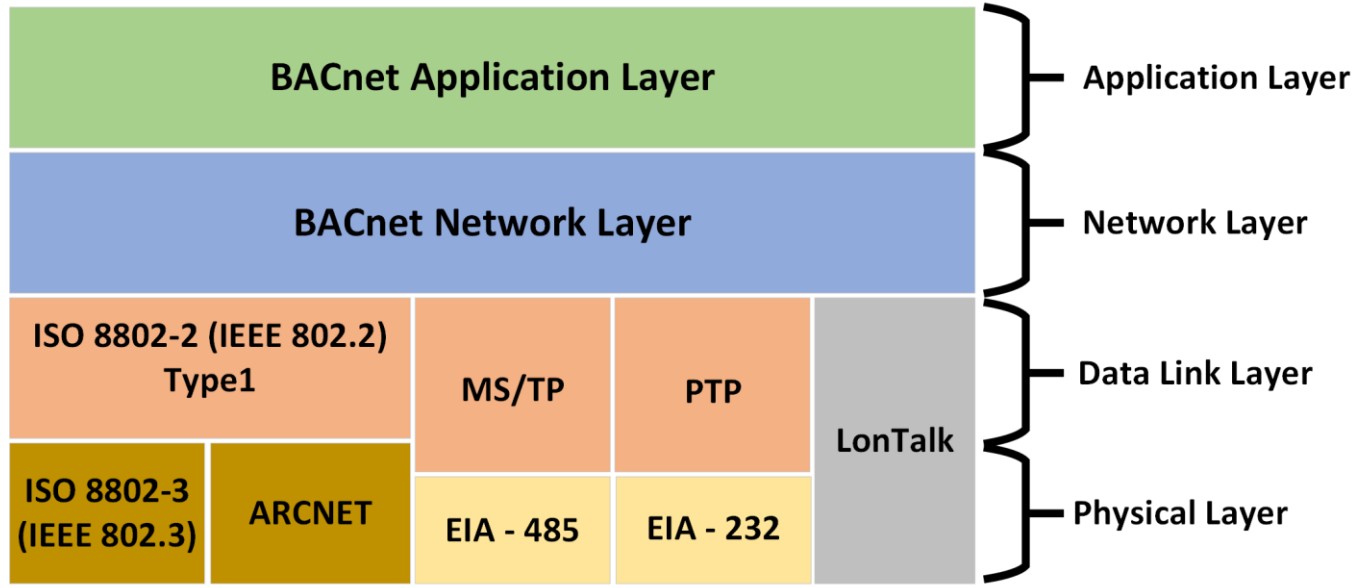

**Figure 11.** Layers of BACnet protocol for building automation and control network.

## 4. Networked Microgrid Cybersecurity

Integrating computing and communication characteristics with the power grid has resulted in several vulnerabilities in the cyber-physical system. These issues represent significant hazards to the physical framework, economy, and society [132]. Several government and private sector assessments have stated that cyber attackers constitute a substantial danger to the security operation of smart grids. Based on the Global State of Information Security Survey in 2015, the energy industry has continuously reported many events to ICS-CERT, with 79 occurrences in 2014 and 46 incidents in 2015, the second-highest number of recorded incidents per sector [133]. It is necessary to develop methods and techniques to detect cyber intrusions and lower their effects to identify and mitigate cyberattacks in smart grids. Table 6 presents a comprehensive overview of cyberattacks on smart grids, providing a concise description of various attack.

**Table 6.** Overview of cyberattacks on smart grids.

| Attack Type | Description | Target | Threat Size | Impact |
|---|---|---|---|---|
| Jamming Attacks | Intentional interference with wireless communication signals renders them unusable | Wireless communication | Small to large | Loss of service, decreased productivity, revenue loss |
| Man-in-the-Middle Attacks | Intercepting and manipulating communication between two parties without their knowledge | Wireless or wired networks | Small to large | Loss of service, decreased productivity, revenue loss |
| False Data Injection | Injecting false or fabricated data into a communication stream or database | Internet of Things (IoT) | Small to large | Stolen credentials, data theft, identity theft |
| Spoofing Attacks | Faking the identity of a person, device, or network to gain unauthorized access or mislead a target | Computer or network | Small to medium | Unauthorized access, data theft or manipulation, system disruption |
| Flooding Attacks | Overwhelming a network or system with traffic, rendering it unusable or causing it to crash | Website or network | Small to large | Service downtime, network congestion, loss of data |

**Table 6.** *Cont.*

| Attack Type | Description | Target | Threat Size | Impact |
|---|---|---|---|---|
| Puppet Attacks | Malware that infects and takes control of a targeted device or network, turning it into a puppet that the attacker can remotely control | IoT or network | Small to large | Unauthorized access, data theft or manipulation, system disruption |
| Masquerade Attacks | Pretending to be a legitimate user, device, or application to gain unauthorized access or privileges | Computer or network | Small to medium | Unauthorized access, data theft or manipulation, system disruption |
| Distributed Denial-of-Service Attacks | Coordinating a large number of devices or systems to flood a target with traffic, overwhelming its ability to respond and rendering it unusable | Website or network | Large | Service downtime, network congestion, loss of data |

### 4.1. Cyberattacks in Smart Grids

- Jamming Attacks

Jamming attacks involve the intentional emission of radio signals that interfere with transceivers to obstruct data transmission and reception [134]. Unlike radio frequency interference (RFI), which is unintentional and brought on by nearby transmitters using the same or similar frequencies, this is not interference. RFI can happen when several wireless sensor networks (WSNs) in a specific area share the same frequency channel. In contrast, jamming attacks are specifically intended to interfere with the operation of a particular system or network. Therefore, it is essential to have efficient countermeasures in place to stop such attacks, especially in critical infrastructures such as smart grids, and to minimize their effects.

- Man-in-The-Middle Attacks

One of the most well-known attacks in computer security is the man-in-the-middle (MITM) attack, a top concern for security experts. MITM focuses on the data that travel between endpoints and the data's confidentiality and integrity [135]. The MITM attacker can intercept, modify, replace, or change the communication traffic of the target victims (this sets an MITM apart from a straightforward eavesdropper). Furthermore, victims think that the communication channel is secure because they are unaware of the intrusion. Different communication channels can carry out MITM attacks, including GSM, UMTS, LTE, Bluetooth, Near-Field Communication (NFC), and Wi-Fi. The attack targets include the confidentiality and integrity of the data and the data that flow between endpoints [136].

Attacks using MITM seek to:

1. Eavesdrop on a conversation violates confidentiality.
2. Compromise integrity through communication interception and message modification.
3. Force one of the parties to stop communicating by intercepting and eradicating messages or altering messages, interfering with availability.

- False Data Injection Attacks

False data injection attacks are significant cyberattacks that target the smart grid's wide-area measuring and monitoring systems. These attacks are designed to tamper with readings from several power grid sensors and phasor measuring units to deceive the operation and control centers. False data injection attacks can be hazardous to networked microgrid systems because they can affect the precision of measurements and control signals across different microgrids. An attacker can introduce false data into a microgrid's sensors and control systems and spread to other connected microgrids, causing extensive disruption and system failure [137–139].

- Spoofing Attacks

A spoofing attack is one of the vulnerabilities in wireless networks, which occurs when an intruder successfully pretends to be legitimate. These attacks will degrade network performance by violating network protocols. A spoofing attack occurs when a malicious

party impersonates another network device or user to launch attacks against network hosts, steal data, spread malware, or circumvent access controls [140].

- Flooding Attacks

A flooding attack is a sort of denial-of-service (DoS) attack in which an attacker delivers an enormous number of messages or requests to a target network or computer to overflow its resources and make it inaccessible or useless. The target's resources in a flooding assault could be the network bandwidth, network channels, switch or router buffers, or CPU cycles of the sender and recipient of time-critical messages [141]. A flooding attack can negatively impact the delivery of time-critical messages since the attacker's flood of messages may cause legitimate time-critical messages to be dropped or delayed, interrupting the intended service or communication.

- Puppet Attacks

A new DoS attack known as the puppet attack can cause a denial of service in the AMI (advanced metering infrastructure) network. Any normal node can be chosen by the intruder and used as a puppet node for attack packets. The attacker will take control of the puppet node when it receives these attack packets and will flood it with further packets to deplete the node's energy and the network's communication bandwidth [142].

- Masquerade Attacks

Masquerade assaults are a cyberattack in which an attacker impersonates a genuine user or entity to access systems containing sensitive data or to conduct other destructive operations [143]. The risk associated with masquerade assaults comes from the attacker's ability to conceal their true identity, which allows them to get around authentication and access constraints that would typically bar unauthorized access to a system or network. Once the attacker has access to the system or network, they could be able to execute additional attacks, such as data theft, file modification or deletion, or malware installation, while still pretending to be a legitimate user.

- Distributed Denial-Of-Service Attack

Distributed denial-of-service (DDoS) attacks are a typical sort of danger to the smart grid that seriously jeopardizes the availability of resources in communication networks. Any occurrence known as a DDoS assault prevents legitimate users from accessing resources, which might reduce or stop the network from operating as it should [144]. Therefore, DDoS assaults may also result in network collapse, power outages and overloads in the smart grid, or even a severe accident [145]. A distributed denial of service (DDoS) attack on a microgrid or networked microgrid network can be directed at any system–system node, such as a smart meter, an aggregator, or the headend.

DDoS attacks can be conducted in several ways, such as:

1. Network layer attacks involve flooding the target system with traffic to target the network bandwidth and infrastructure.
2. Transport layer attacks target network protocols such as TCP or UDP by exploiting flaws in how packets are transmitted.
3. Application layer attacks involve sending many requests to a specific application or service, such as a web server or a database, that the server cannot manage.

### 4.2. Cyberattack Vulnerabilities in Networked Microgrids

Networked microgrids are susceptible to cyberattacks that can cause operational disruptions, malfunctions, or even power outages, just like any other connected system. Unauthorized access, data breaches, and cyberattacks are three common vulnerabilities that pose severe risks to networked microgrids. Unauthorized access describes an event in which an attacker successfully exploits weaknesses in the Operating Systems (OSs), software, or hardware of a microgrid's components to gain unauthorized access to its systems and networks. Weak passwords, open communication channels, and unpatched

software vulnerabilities can all be used by attackers to gain access. Attackers may take information, interfere with operations, or even harm the microgrid directly when they have gained access. Unauthorized access can cause power outages, equipment damage, and operational disruptions, costing a company a lot of money and harming its reputation. A further vulnerability that could compromise the privacy of networked microgrids is a data breach. Data breaches happen when intruders penetrate the microgrid's data storage servers or network and take sensitive information, including customer, financial, or operational data. Data breaches can be expensive for microgrid operators and users since they can result in regulatory fines, financial losses, and reputational harm. Another critical area of danger for networked microgrids is cyberattacks. Cyberattacks can come in various shapes and sizes, including ransomware, spyware, and denial-of-service attacks. Denial-of-service attacks can overwhelm the microgrid's systems, which would then break down and result in operational issues and power outages. The microgrid's systems are vulnerable to malware attacks, which give hackers access to control how it functions, steal data, and even harm the microgrid physically. Data on the microgrid may be encrypted by ransomware attacks, rendering it inaccessible unless a ransom is paid. Cyberattacks have the potential to cause significant financial losses. Networked microgrids are particularly vulnerable to insider attacks by employees who purposely or accidentally breach the system's security. Identifying these people's malicious intent may be challenging because they have valid access to the microgrid and could know how it functions. Unintentional insiders can unintentionally trigger safety hazards through human mistakes or negligence, while nefarious insiders may steal valuable information or impair the microgrid's operations. Implementing reasonable access restrictions, evaluating systems, and regular employee training on relevant security policies and procedures are necessary for identifying and managing insider threats.

### 4.3. Techniques for Detecting Cyberattacks on Smart Grids

The detection of cyberattacks on smart grids is possible using various techniques. Detecting anomalies, network traffic analysis, log analysis, and intrusion detection systems (IDSs) are a few of the often-employed techniques. Most of them are demonstrated in the following section.

- Filtering-Based Techniques

Cyberattack detection in smart grids frequently uses filtering-based algorithms. Threshold-based and bloom filtering are the two main methods included in this group. Although simple to create, threshold-based techniques could be more effective due to their limitations. On the other hand, because of their minimal memory and processing power requirements, bloom filters can efficiently detect anomalies in a smart grid system. Studies have demonstrated the effectiveness of bloom filters in spotting flooding attacks against SCADA systems and signaling protocols, with one study [146] obtaining a 97% accuracy rate. The ease of use, but potential inefficiency and expense, of filtering techniques has been contrasted with that of other detection techniques in other research that looked at social engineering attempts. Overall, filtering-based approaches, especially bloom filters, are well known for their small size. In [147], the authors compared filtering techniques to other detection methods for detecting social engineering attacks. They emphasized that while filtering techniques are simple, they must be more extensive and costly. However, filtering techniques such as bloom filters are considered space-efficient and valuable in specific situations within a smart grid.

- Intrusion Detection System

Cyberattacks on the smart grid infrastructure can be found using intrusion detection systems (IDSs), which are widely acknowledged [148]. These systems are equipped with the capacity to review and evaluate security information to spot potential malicious threats. One significant advantage of an IDS is its ability to recognize undetected or zero-day attacks.

In networked DC microgrids, the process of detecting and responding to hostile behaviors that endanger the system's security is known as cyber intrusion detection. It entails employing intrusion detection systems to examine system events and network traffic to find potential security holes. In [149], the authors describe a distributed optimal dynamic state estimation technique for detecting cyber intrusions in networked DC microgrids. The suggested solution employs a consensus-based algorithm to assess the system's state and detect cyber breaches. The results reveal that the recommended technique accurately detects cyber intrusions in networked DC microgrids. In [150], the authors suggest using bloom filters to stop flooding attacks against Session Initiation Protocol (SIP)-based services. The study demonstrates that bloom filters might be a powerful tool for identifying and thwarting such attempts.

In [151], the authors describe the hybrid-multilevel anomaly prediction approach, HML-IDS, for SCADA system intrusion detection. The suggested method identifies real-time anomalies by combining statistical analysis, clustering, and machine learning algorithms. The study's findings demonstrated that HML-IDS performed better than other current methods for SCADA system intrusion detection, achieving high accuracy and low false-positive rates. In [152], the authors suggest a distributed intrusion detection system (FDIDS) based on fog to identify false-metering attacks in smart grid systems. The suggested system uses a distributed architecture of fog nodes to identify and stop malicious activities in the network. The findings demonstrate that the recommended system performs better than traditional centralized intrusion detection systems regarding the detection accuracy and response time.

- Prediction Models

Prediction models are mathematical models that forecast future occurrences or trends using statistical analysis and artificial intelligence techniques. Prediction models are frequently used to predict and spot attacks before they occur, such as cyberattacks on smart grids. These models can be developed with historical data and discover potential attacks by analyzing data patterns.

Prediction models come in various forms that can be applied to the security of smart grids. Rule-based models, for instance, look for potential attacks using several predetermined rules. These rules may be based on variables such as the nature of network traffic, its origin, and the time of day. Machine learning models are different prediction models that employ algorithms to identify patterns in past data and forecast future events. Other machine learning methods, such as supervised or unsupervised, can be used to train these models. For instance, a novel evolutionary-algorithm-based method for cyberattack detection is presented in [153]. The proposed approach emphasizes identifying the characteristics of cyberattacks on the energy system, including shifts in energy usage and unusual network traffic. After selecting the most pertinent features for identifying signs of attacks using evolutionary algorithms, the authors trained a classifier based on machine learning using these features. The study's findings demonstrated that the suggested technique could accurately and successfully detect cyberattacks on the power grid. Another study [154] developed a method for identifying cyber assaults in smart grids by combining supervised learning with heuristic feature selection. The strategy was to choose essential data from the dataset and apply a classifier to anticipate attack events. The approach was shown to be successful in identifying various forms of assaults, according to the results.

Prediction models such as ARMA and ARIMA have been put forth for spotting cyberattacks in networks of smart grids. These models forecast future behavior using time-series data and historical data. For example, in [155], the authors demonstrate a novel approach for anomaly detection and online forecasting that utilizes the ARIMA model. The approach is intended to be effective and flexible and can deal with both regular and abnormal data. The method was assessed by the authors using a variety of datasets from the real world, and they demonstrated that it could produce innovative results.

- AI-Based Techniques

Artificial intelligence (AI)-based techniques are a large group of approaches that utilize AI to solve issues. The field of artificial intelligence (AI) is concerned with the development of intelligent agents, or autonomous reasoning, learning, and acting systems. An effective tool for identifying cyberattacks is machine learning. It is more scalable and customizable than conventional methods of cyberattack detection. ML techniques can be divided into three categories: unsupervised, semi-supervised, and supervised.

Unsupervised approaches involve discovering patterns, frameworks, or information within unlabeled data. In semi-supervised methods, some data are labeled based on professional knowledge during the data acquisition phase. Finally, in supervised approaches, the entire dataset is labeled using professional learning, and a function or model is found that adequately explains the data [156]. Using ML techniques to track and detect cyberattacks within smart grid systems has received the attention of various researchers. For instance, the authors of [157] present a hybrid deep learning approach to detect false data injection attacks. This was achieved by combining a deep learning network (DNN) with an encoder that identifies the subset of measurements affected by FDIA. The authors concluded that the suggested approach improves the security of the smart grid. In [158], the authors propose a technique for identifying DoS and DDoS attacks on Internet of Things (IoT) devices using a deep learning approach called ResNet, which is a type of convolutional neural network (CNN) that is frequently employed for image recognition tasks. The paper concludes that the suggested approach may be reliable for identifying DoS and DDoS assaults on IoT devices, enhancing the security of IoT systems.

The method described in [159] uses machine learning to identify temporal synchronization attacks on cyber-physical systems (CPSs). For CPSs to function well and dependably, time synchronization is essential. Time synchronization attacks may stop a CPS from working, harming the system. The suggested solution detects temporal synchronization attacks with high recall and precision using Random Forests (RFs) and Support Vector Machines (SVMs). The study concludes that this technique is dependable for identifying temporal synchronization threats in CPSs and may be applied to enhance CPS security. The authors of [160] propose an anomaly intrusion detection system for smart grids using ensemble learning methods. They used algorithms such as Random Forest, AdaBoost, and Bagging to create a classifier to identify abnormal events and detect intrusions. Another example in [161] presents a physical-layer intrusion detection and location system for smart grids using machine learning techniques. They use a neural network to detect and locate the intrusion's source by analyzing the smart grid's physical-layer signals. In [162], a method is proposed for detecting and preventing man-in-the-middle (MitM) spoofing attacks in MANETs (mobile ad hoc networks) using predictive techniques in artificial neural networks (ANNs). The authors use an ANN to predict the trustworthiness of nodes in the network and detect MitM attacks. Overall, the paper demonstrates the potential of machine learning in improving the security of smart grids and MANETs.

A novel method for identifying covert cyberattacks in smart grid networks is explained in [163]. The plan is predicated on highly randomized trees (ERTs), a machine learning technique well renowned for recognizing intricate patterns in data. The ERTs identify whether new traffic is regular or an attack after training on a dataset of attack and regular traffic. The method was assessed using a real-world dataset of traffic on smart grids, and the results demonstrate that it is highly accurate at spotting covert cyberattacks.

- Localization-Based Techniques

In the literature, several localization methods have been put forth, such as Received Signal Strength (RSS), Received Signal Strength Indicator (RSSI), Time Difference of Arrival (TDoA), and Angle of Arrival (AoA). RSS-based methods are frequently utilized in communication technologies for some functions, including transmitter location detection. One example is a spoofing attack detection method that uses spatial correlation features extracted from RSS stream data, as described in [164]. The method uses the ratio of out-

of-bound frames and the Summation of Detailed Coefficients (SDCs) in the Discrete Haar Wavelet Transform (DHWT), two distinguishing characteristics of RSS streams. The technique has an excellent spoofing attack detection rate, but its poor localization accuracy is a significant drawback. Another strategy suggested in [165] reduces data redundancy by using RSS stream features but emphasizes the RSS-based technique's poor accuracy. Although they have this drawback, RSS-based techniques are still practical for spotting network spoofing attempts. In [166], the authors suggest a localization method for wireless sensor networks called Angle of Arrival (AoA). The authors explain how AoA can infer a node's location from the angle of the signal it receives from various reference nodes. The suggested method is based on a triangulation method that locates the target node by measuring the angles between the two reference nodes and the target node. The proposed technique can attain a localization accuracy of a couple of feet with a minimal number of reference nodes, according to experiments conducted by the authors to assess its accuracy. The outcomes demonstrate that the suggested method may be helpful for localization in wireless sensor networks.

*4.4. Techniques for Mitigating Cyberattacks on Smart Grids*

One of the crucial axes that the researchers investigated is techniques for mitigating cyberattacks or countermeasures. Various classifications have been presented to address security threats in smart grids. The authors of [167] proposed a classification scheme that divides countermeasures into four categories: cryptographic functions, classification algorithms, personnel identification, and channel characteristics. In this section, we categorize countermeasures in smart grid networks into two main groups, prevention-based and protection-based, as shown in Figure 12.

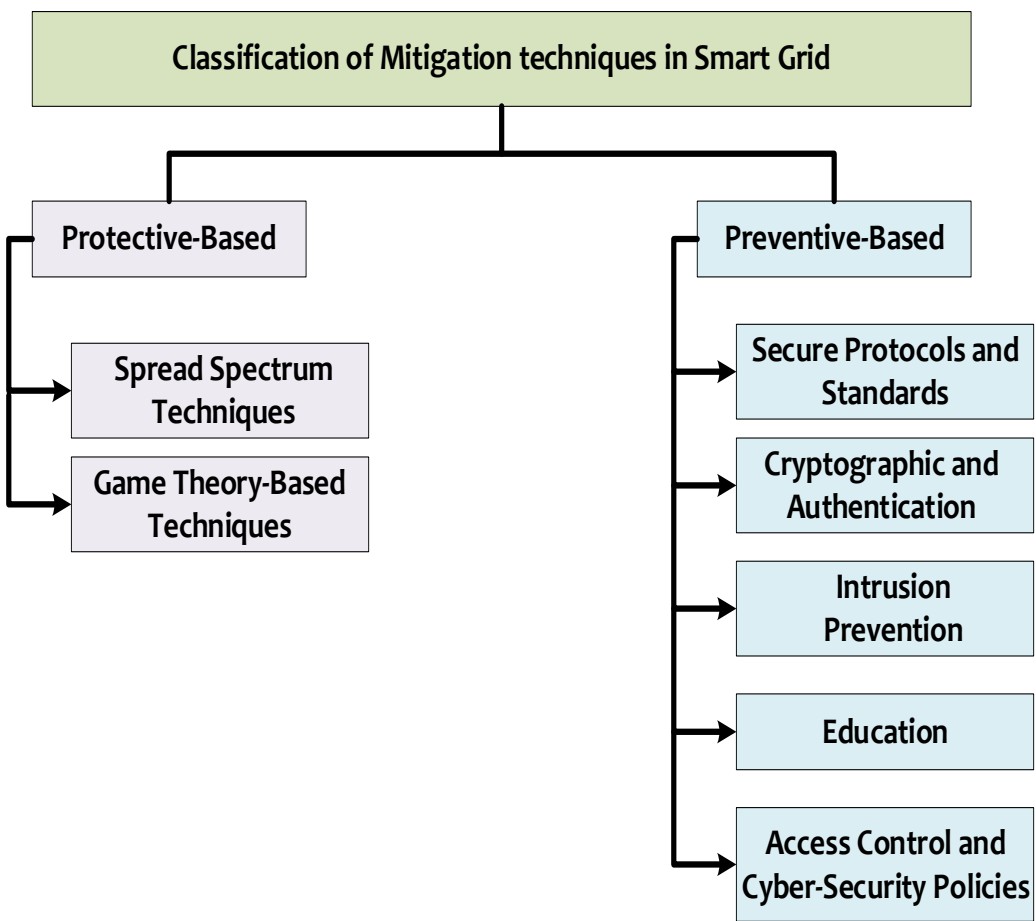

**Figure 12.** Classification of mitigation techniques.

Computer-based countermeasures can be divided into five categories: secure protocols and standards, cryptographic functions, intrusion preventions, security training, and access control and cybersecurity policies. Two categories of non-computerized countermeasures are (1) education and (2) access control and cybersecurity regulations.

- Prevention-Based Mitigation techniques

  1. Secure Protocols and Standards Secure protocols are crucial for smart grid networks to transmit data securely and accurately. Some examples of these protocols include secure DNP3, IPsec, TLS, and SSL. In contrast to TLS and SSL, which offer authentication and encryption at the transport layer, IPsec offers both at the network layer. Numerous applications, such as online transactions and remote access VPNs, use these protocols regularly. One of the most popular protocols for facilitating interactions among remote terminal units (RTUs) and control centers in the context of the smart grid is DNP3 (Distributed Network Protocol). However, by default, DNP3 lacks any security features, which leaves it open to intrusions. As a result, secure DNP3 became available to offer message integrity protection, authentication, and encryption. Secure DNP3 incorporates security features, including user identification, message authentication, encryption, and key management, to defend against various incidents, such as replay attacks, man-in-the-middle attacks, and eavesdropping [168]. Various security protocols were studied to enhance the security and integrity of data transmission in smart grid systems. For example, the IEEE 802.11i protocol and smart grid secure protocol (SGSP) were proposed in [169–171]. These protocols include data integrity, confidentiality, and additional authentication.

  2. Cryptographic and Authentication Smart grid networks commonly use cryptographic and authentication methods as defenses against cyberattacks. Cryptography provides secure communication and confidentiality by encrypting sensitive data and restricting access. Additionally, it guarantees integrity and authenticity by using digital signatures and hashing algorithms to check the accuracy of the data and confirm the sender's identity. Authentication mechanisms are essential for verifying consumer and device identities and ensuring that only authorized parties can access sensitive systems and data.

  3. Intrusion Prevention It is crucial to stop or eliminate illegal activity in the network to improve the performance of smart grids. Using firewalls and antivirus software is a classic method for achieving this. Although they are not a perfect solution, firewalls and antivirus software are crucial tools for safeguarding against attacks on smart grids. Utilizing additional security measures, such as intrusion detection and prevention systems (IDS/IPS), is advised. In [172], firewalls are defined as either hardware or software systems that can keep an eye on network activity using a variety of protocols. Firewalls and antivirus software, however, cannot stop complex or unknown cyberattacks. Other security measures have been suggested for this purpose, including network data loss prevention (DLP), intrusion prevention systems (IPSs), security information and event management systems (SIEMs), file integrity monitoring (FIM), and automated security compliance, to lessen or prevent the effects of cyberattacks on the network. Data loss prevention (DLP) is a method of protection that monitors and regulates the transfer of data within a company's network to detect and stop data breaches. Data transfers, messages, and other network activities can all be watched by DLP systems to identify and stop sensitive data from exiting the network, whether on purpose or accidentally. An IPS, a network security technology, performs the real-time detection and prevention of network intrusions. It is an advanced intrusion detection system (IDS) that recognizes malicious network activity but does nothing to stop it. By obstructing network traffic or discarding malicious packets, an IPS, on the other hand, can identify and halt malicious network activity. A security technology called SIEM gathers and analyzes security events from

various sources to provide an overview of a company's security posture. SIEM can collect data from network servers and other security systems. Moreover, FIM is a security method to spot unauthorized alterations to essential files and system settings. To spot any unauthorized activity, FIM systems can monitor file additions, deletions, changes, and access permissions. This aids in detecting and preventing cyberattacks that target the integrity of files and systems.

4.  Security Training/Education Security and technical countermeasures are only sufficient to assure complete protection for the system or smart grid if periodic training and education are provided to employees and customers, which can be crucial in preventing cyberattacks. Most companies, utilities, and universities require crews and employees to take cyberattack awareness training as it is crucial in preventing cyberattacks and ensuring complete protection for the system or smart grid. The authors of [173] recommend that utility companies provide security courses or programs to their employees. These training sessions cover social engineering, phishing, password management, and the value of updating firmware and software.

5.  Access Control and Cybersecurity Policies Several efficient tactics, such as attribute-based or policy-based access control, can control access privileges in smart grid networks. Employees who have been permitted to do so define the authorization policies that other employees and users must follow to receive approval, thereby preventing physical or digital attacks.

- Protection-Based Mitigation techniques

1.  Spread-Spectrum Techniques In order to defend against jamming attacks, smart grid networks frequently employ the spread-spectrum techniques known as frequency-hopping spread spectrum (FHSS) and direct sequence spread spectrum (DSSS). FHSS abruptly shifts a signal's transmission frequency across several channels, whereas DSSS multiplies the signal with a noise-like sign to spread it across a larger bandwidth. Both methods ensure that jamming attacks cannot easily disrupt or intercept the transmission signal. The FHSS technique is suggested in [174,175] as a defense against jamming and collision attacks on the smart grid network. This method employs a broader bandwidth than a single carrier frequency to ensure security. The authors of [176] assert that FHSS techniques have several benefits, including efficient management of the multipath effect.

2.  Game-Theory-Based Techniques Mathematical models incorporating strategic decision-making interactions are known as game-theory-based techniques. Game theory has been applied to the development of false data injection attack prevention methods in the framework of smart grid security. The authors of [177] proposed a two-layer game theory method for optimizing the allocation of different defense resources using information from multiple sources. A cost-effective and efficient approach for large-scale power systems and the infrastructure of the smart grid was provided by another study [178], which suggested a game theory model depending on the minimax regret method.

### 4.5. Policies and Awareness Training

Mitigating cyberattacks on power grids necessitates efficient cyber techniques for detecting and mitigating attacks. However, human behavior also plays a role in the success of these attacks, demonstrating that technological solutions alone cannot ensure security [179]. Kaspersky Lab published a report [180] concluding that 52% of companies state that employees are the most significant weakness in cybersecurity. Even though incidents caused by malicious attacks are more expensive, human error is still considered costly, with an average cost of USD 2.94 million [181]. As a result, it is critical to educate employees about social engineering techniques and equip them with the skills to identify and respond to security incidents, including training, awareness courses, and simulation scenarios. Training

exercises should include password hygiene, safe communication practices, spotting and reporting suspicious activity, and proper use of business resources, such as email systems and file-sharing platforms. Employees can acquire the cybersecurity skills required to efficiently handle and address dangers and threats to their private data. Employees will be better able to manage cybersecurity responsibilities, as they will be more aware of the risks and hazards that businesses face in cybersecurity [182].

Employees with direct access to critical equipment should be the focus of awareness training. Additionally, it is important to adapt awareness training to the constantly changing cyberattack landscape, in addition to creating and maintaining actionable incident response plans and guidelines that give employees specific instructions on responding to security incidents [183,184].

The response to cybersecurity incidents should be regularly practiced using an appropriate training simulator that closely resembles the typical process control networks seen in power grids. In general, there is a need for a comprehensive strategy for cybersecurity that includes both technical solutions and policies, procedures, and awareness [185].

## 5. Conclusions

This paper extensively reviews current research on networked microgrids (NMGs), examining various aspects, such as their architecture, control systems, protection mechanisms, economics, communication methods, and cybersecurity considerations. The findings unequivocally demonstrate the potential of NMGs as a transformative technology for the future of power grids. NMGs offer many benefits, including improved dependability, enhanced grid resilience, and increased sustainability, thereby contributing to the overall stability and reliability of the power grid system. However, the successful deployment of NMGs on a larger scale necessitates addressing key challenges, such as developing advanced communication protocols, implementing robust cybersecurity measures, and establishing appropriate legislation and standards frameworks. This comprehensive review underscores the critical need for continued research, innovation, and collaboration among researchers, industry professionals, and policymakers to ensure the secure and effective implementation of NMGs. It provides valuable insights for individuals interested in the development, execution, and utilization of NMG systems, highlighting the significance of ongoing advancements in this rapidly evolving field.

**Funding:** This research received no external funding.

**Conflicts of Interest:** The authors declare that there are no conflicts of interest regarding the publication of this paper.

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
