# Peer review of "A Comprehensive Review of Architecture, Communication, and Cybersecurity in Networked Microgrid Systems"

_inventions, doi:10.3390/inventions8040084_

Round 1

Reviewer 1 Report

In this paper the authors investigate architecture, communication, and cybersecurity issues in networked microgrids. It is a well written paper. However some points should be mentioned and should be included within the manuscript in order to improve the publication.

§  The reasons for writing the paper or the aims of the research should be given in the abstract.

§  The abstract should be more informative about the content of the manuscript. The authors explain the correlation between the following sections in the introduction. It is not necessary to repeat it in the abstract.

§  The authors need to clarify and explain the difference of the current study with the available literature.

§  A better classification and a comparative analysis between the references would be useful.

§  It is suggested that the authors analyse better the section ‘3.7. Policies, Awareness training’.

Reviewer 2 Report

Dear Authors, thank you for your work. Your review is a wide review of microgrids and networked microgrids that includes the Cybersecurity aspects of it. Regarding this paper, I have the following comments:

1.- The paper is too long, 43 pages. Rather than a review, It could be a chapter of a book or its contents could be distributed along the different chapters of a book about Microgrids.

2.- The English grammar is ok, but there are a lot of errors, blank spaces in the text, acronyms defined in not the proper location of the text, and others. I will try to send you a highlighted version. Because listing all of them will be a problem.

3.- I've found many well-known explanations and definitions of concepts like "Latency, Bandwith, Reliability, QoS, Scalability, Interoperability, Security,  Standardization, Ethernet, Optical Fiber Communication, Serial Communication, Cellular Communications, Zigbee, Wi-Fi,..."

This made the document really long.

4.- Though the English grammar is ok, I have found problems reading the document. It didn't catch my attention. The long extension of the paper doesn't help.

5.- The novelty of the review paper regarding other similar review papers about microgrids seems to be the Cybersecurity section. Then, surely to center that review only on this part would be more reasonable.

6.- The review of the paper tries to cover all the aspects of the microgrid. It is a general paper. However, because pretends to be general, I miss dealing with the issue of protections, especially in a multi-network-microgrid case. Protection is a complex issue inside a complex system. Nothing is said about that.

7.- The listing of all the standards related to microgrids is a good point of the paper.

8.- Here below, I point to you, a few grammar bugs I've found in the paper. There are more highlighted in the uploaded file. Please correct them and fix the other ones still existing in the paper.

9.- In line 288 appears the following statement "Droop control is widely used for power sharing between different resources. However, it has many disadvantages, including usage limitations with nonlinear loads, instability issues, and low transient performance [51]."

There are these recent contributions that you didn't consider referencing and that overcome the problem related to nonlinear loads when using the Droop method. Please include them on the reference list:

  1. [1]  Li, M.; Matas, J.; Mariachet, J.E.; Branco, C.G.C.; Guerrero, J.M. A Fast Power Calculation Algorithm for Three-Phase Droop-Controlled-Inverters Using Combined SOGI Filters and Considering Nonlinear Loads. Energies 202215, 7360, doi.org/10.3390/en15197360.
  2. [2]  El Mariachet, J.; Guan, Y.; Matas, J.; Martín, H.; Li, M.; Guerrero, J.M. HIL-assessed fast and accurate single-phase power calculation algorithm for voltage source inverters supplying to high total demand distortion nonlinear loads. Electronics 20209, 1643, doi.org/10.3390/electronics9101643.
  3. [3]  El Mariachet, J.; Matas, J.; Martín, H.; Li, M.; Guan, Y.; Guerrero, J.M. A power calculation algorithm for single-phase droop-operated-inverters considering linear and nonlinear loads HIL-assessed. Electronics 20198, 1366, doi.org/10.3390/electronics8111366

-   Line 13, The present study comprehensively investigates architecture, Communication, and cybersecurity issues in NMGs. (why Communication appears with a capital letter?

-  The same happens with the word “Challenges”, one line below the previous one

- In the abstract, you explain what is going to be in section I, II, and III.  This is not normal. Usually is performed at the of the Introduction section.

- Line 51, remove the adjective “typically”. It could be, but is not a general trend.

- Line 37, here appears a listing In the text as (1) ….. (2)…….(3)…….(4)……….(5). Please, list that item by item, i.e.,

              - (1) ……………

              - (2) ………….

              - (5) ………………

-  Line 105, don’t you think that the statements made in this text are the same?: “by exchanging excess energy and resources with other connected microgrids by exchanging energy and resources among interconnected microgrids,”

-  Table 1, in standard IEEE 1547.7 (2013) appears “DREs” and should be DERs

-   Line 455: “distributed energy resources (DERs)”. Please, this acronym should appear at the beginning of the first time. Please, check that question because it could happen also for other acronyms.

-   Line 632: should be “Communication”

-   Line 673: “making the system the system to become unstable.”….”the system appears twice”

-   Line 705, appears two points

-  Line 891, “andhe”

-  Line 913, “microgridsnd”

-  Line 927-928, there is a break in the phrase.

-  Line 939: “containingensitive”

-  Line 962 – 963, there is the head name of Table 5, and it is broken by the introduction of section 3.4

-  Line 1073, “The”

- Line 1140, “The”

- Line 1151, “VVarious”

The English grammar to me is ok, but new a lot of corrections. I indicated some of them to the Authors.

Reviewer 3 Report

The manuscript presents an overview of the problems and challenges associated with implementing microgrids. In this sense, the issues and concepts related to architecture, communication and cyber security in microgrids are discussed in detail. The topic of the manuscript is relevant and useful. My main observations and comments are as follows:

- it would be useful in the final version of the manuscript to discuss the technological and economic aspect of the application of microgrids and to compare the different topologies and concepts in relation to their practical implementation, investments, price advantages or disadvantages;

- there is a lack of in-depth analysis and commentary on the presented concepts and topologies of microgrids. In this sense, a discussion section is missing, and the conclusion section repeats commonly known facts about microgrids;

- I recommend to carefully review all the figures and improve their quality. Moreover, those figures which are taken readily from the relevant literary sources should be cited.

Reviewer 4 Report

What is the difference between networked microgrid, clustering microgrid and multi-energy microgrid. Discuss this in terms of their definition, operation, and advantages and disadvantages.

Suggestion to include tabular data for figure 3 also discussing its advantages and disadvantages as discussed in the literature. Also, need references for each proposed topology if possible.

“maximize global advantages” need more elaboration.

This paper only discusses layer-type control architecture/structure (Hierarchical control). What could be other control architecture/structure(such as Master-Slave, etc) available in the literature? Also, discuss their advantages and disadvantages in tabular form

Suggestion: include advantages and disadvantages in table 4. Also, compare with respect to topology and control method used. Also, discuss how to overcome the disadvantages of each communication technologies. 

Compare each cyber attack type with respect to the topology and control method used. Also, discuss how to overcome with respect to each 

Quality of english is medium and needs minor editing in the manuscript

Round 2

Reviewer 2 Report

I have no more comments

The English grammar seems ok.

Author Response

Thank you for taking the time to review our paper once again. We appreciate your thorough evaluation and consideration of our work. We are pleased to note that you have no further comments or suggestions regarding the paper. Your positive feedback indicates that the revisions we made have successfully addressed the previous comments and improved the quality of the paper.

We are grateful for your time and valuable input throughout the review process. Your expertise and insights have been invaluable in refining our work. We look forward to the next steps in the publication process.

Thank you once again for your time and consideration.

Reviewer 3 Report

The authors have reworked the manual and reflected the main part of my remarks and comments. I have no further notes.

Author Response

We sincerely appreciate your review and the positive feedback you provided. Thank you for acknowledging the efforts we made to address your previous remarks and comments. We are glad to hear that the revisions we implemented in the manuscript have effectively incorporated your suggestions.  

We are grateful for your time and valuable input throughout the review process. Your feedback has been instrumental in enhancing the quality and clarity of our paper. Your final confirmation that you have no further notes or comments is reassuring.

Once again, we extend our gratitude for your thorough evaluation and valuable contributions to our work. We look forward to the next steps in the publication process.

Thank you for your time and consideration.

Reviewer 4 Report

just improve the analysis of multiple control structures, including Centralized, Decentralized, and Distributed with clear and separate sub-headings

nor required

Author Response

Thank you for your valuable feedback and suggestions regarding the analysis of multiple control structures in our paper. We have taken your comment into consideration and made the necessary revisions to improve the clarity and organization of this section. 

We are grateful for your time and valuable input throughout the review process. Your feedback has been instrumental in enhancing the quality and clarity of our paper. Your final confirmation that you have no further notes or comments is reassuring.

Once again, we extend our gratitude for your thorough evaluation and valuable contributions to our work. We look forward to the next steps in the publication process.

Thank you for your time and consideration.